# A flux tower site attribute dataset intended for land surface modeling

Jiahao Shi[1], Hua Yuan[1], Wanyi Lin[1], Wenzong Dong[1], Hongbin Liang[1], Zhuo Liu[1], Jianxin Zeng[1], Haolin Zhang[1], Nan Wei[1], Zhongwang Wei[1], Shupeng Zhang[1], Shaofeng Liu[1], Xingjie Lu[1], Yongjiu Dai[1]

[1]Southern Marine Science and Engineering Guangdong Laboratory (Zhuhai), Guangdong Province Key Laboratory for Climate Change and Natural Disaster Studies, School of Atmospheric Sciences, Sun Yat-Sen University, Zhuhai 519082, China

*Correspondence to*: Hua Yuan (yuanh25@mail.sysu.edu.cn)

**Abstract.** Land surface models (LSMs) require reliable forcing, validation, and surface attribute data as the foundation for effective model development and improvement. Eddy covariance flux tower data are widely regarded as the benchmark for LSMs. However, currently available flux tower datasets often require multiple aspects of processing to ensure data quality before application to LSMs. More importantly, these datasets frequently lack site-observed attribute data, such as fractional vegetation cover and leaf area index, which limits their utility as benchmarking data. Here, we conducted a comprehensive quality screening of the existing reprocessed flux tower dataset, including the proportion of gap-filled data, energy balance closure (EBC), and external disturbances such as irrigation and deforestation, leading to 90 high-quality sites. For these sites, we collected vegetation, soil, and topography data, as well as wind speed, temperature, and humidity measurement heights from literature, regional networks, and Biological, Ancillary, Disturbance, and Metadata (BADM) files. We then compiled the final flux tower attribute dataset by filling in missing attributes with global data and classifying plant functional types (PFTs). This dataset is provided in NetCDF format with necessary descriptions and reference sources. Model simulations revealed substantial disparities in output between the attribute data observed at the site and those commonly used by LSMs, underscoring the critical role of site-observed attribute data and increasing the emphasis on flux tower attribute data in the LSM community. The dataset addresses the lack of site attribute to some extent, reduces uncertainty in LSM data source, and aids in diagnosing parameter as well as process deficiencies. The dataset is available at https://doi.org/10.5281/zenodo.12596218 (Shi et al., 2024).

## 1 Introduction

Land surface models (LSMs) simulate the exchange of carbon, water and energy fluxes between soil, vegetation and atmosphere, and are essential tools for comprehending and predicting mass and energy interactions between the earth's biosphere and atmosphere (Pitman, 2003; Williams et al., 2009). The key role of LSMs is to provide the land surface boundary conditions for climate and weather forecast models (Mariotti et al., 2018; Pitman, 2003), as well as uncoupled

stand-alone runs to investigate terrestrial water resources, ecology, and carbon storage (Crow et al., 2012; Humphrey et al., 2021; Ukkola et al., 2016a). Therefore, LSMs offer valuable insights for addressing environmental issues and mitigating climate change. Offline (i.e., uncoupled) LSMs are forced by meteorological data including wind speed, air temperature, specific humidity, air pressure, precipitation, and downward longwave and shortwave radiation. Flux towers measure the

cycling of carbon, water and energy between the biosphere and atmosphere, providing observations with meteorological data that can be used to force offline LSMs. These observations are characterized by high temporal resolution (typically 30 min), continuous observations, direct flux measurements, and often over span years. For these reasons, they are regarded as benchmarking data for LSMs calibration, evaluation, and enhancement, enabling model development from sub-daily to seasonal and interannual scales. Numerous studies have leveraged flux tower data for developing LSMs (Best et al., 2015;

Blyth et al., 2010; Harper et al., 2021; Melton et al., 2020; Stevens et al., 2020; Stöckli et al., 2008; Ukkola et al., 2016b; Zhang et al., 2017). However, despite their significance, flux tower data were not originally designed for testing and validating LSMs. When applied to LSMs, these datasets suffer from poor data quality and a deficiency of site attribute data.

FLUXNET2015 is currently the most widely used flux tower dataset (Pastorello et al., 2020). However substantial preprocessing is frequently required to ensure the reliability of meteorological forcing and flux assessment data for LSM. To

reduce repetitive data processing efforts and improve consistency, Ukkola et al. (2022) integrated three flux tower datasets (FLUXNET2015, La Thuile, and OzFlux) and then performed screening, gap-filling, and other procedures to resolve issues such as missing data and energy balance closure (EBC). This effort resulted in a dataset called PLUMBER2, comprising 170 high-quality sites is tailored for LSMs. This work considered as many available flux tower datasets as possible and used an automated, reproducible data screening process. However, the PLUMBER2 dataset only performs quality checks on

meteorological forcing data, not on flux assessment data, to obtain more available years of data and enable models to be assessed against specific weather and climate events. Consequently, a large proportion of gap-filled flux data is present at some sites. Land surface modelers typically employ stringent quality control procedures to avoid misleading model evaluation results (Blyth et al., 2010; Li et al., 2019; Purdy et al., 2016). Therefore, these existing gap-filled data still require further processing.

Most importantly, these flux tower datasets lack site-observed vegetation, soil, and topography data such as fractional vegetation cover (FVC), Leaf Area Index (LAI), soil texture, slope and aspect. For regional and single-point modeling, the current practice usually involves obtaining these attribute data for LSMs through the inversion of global satellite observations. This approach introduces additional uncertainty into LSMs and diminishes the utility of flux tower data as benchmarking data for model evaluation.

Uncertainty in vegetation and soil data constitutes a significant source of uncertainty in LSMs (Dai et al., 2019b; Li et al., 2018). Vegetation composition and density play a prominent role in modulating the surface energy budget (Bagley et al., 2017; Williams and Torn, 2015), by altering canopy conductance, aerodynamic properties, and albedo, ultimately affecting water and energy fluxes between the surface and atmosphere (Anderson et al., 2011; Bonan, 2008). Similarly, soil texture directly influences various soil hydrological and thermodynamic parameters, including saturated soil water content and soil

thermal conductivity (Arya and Paris, 1981; Minasny and McBratney, 2007). These parameters have a substantial impact on soil temperature and moisture, as well as the terrestrial carbon and water cycle (Dirmeyer, 2011; Entekhabi et al., 1996). Although recent LSM development has attempted to use site-observed attribute data to reduce uncertainty in model results (Harper et al., 2021; Melton et al., 2020), the data used in these studies are typically limited and not publicly available, making it challenging for other researchers to apply these valuable data. Generally speaking, no flux tower dataset can be used directly in developing LSMs, and they frequently lack the necessary site-observed information about soil, vegetation, and other attributes.

To provide more accurate and reliable flux tower data for LSM modeling and validation, we conducted thorough quality control for the site data based on the PLUMBER2 dataset produced by Ukkola et al. (2022), resulting in a total of 90 sites. Subsequently, we carried out an extensive collection of available flux tower attribute data, drawing from sources such as site-related literature and websites. We further complemented the attributes with global data. As a result, we generated a flux tower dataset that can be directly applied to LSMs and contains essential attribute data. Furthermore, through modeling comparison for the four key attribute variables—percentage of plant functional type (PFT) cover (PCT_PFT), LAI, canopy height, and soil texture—we demonstrate how the outputs differ between site-observed attribute data and the default attribute data employed by an LSM. These results emphasize the non-negligible impact of flux tower attribute data on model simulation and development.

## 2 Data and Methods

### 2.1 Datasets

The data used in this study can be categorized into four groups, as illustrated in Table 1. Firstly, PLUMBER2 serves as the dataset for data quality screening. The second group comprises the attribute sources, including 113 site-related literature, seven flux regional networks, and the Biological, Ancillary, Disturbance, and Metadata (BADM) files provided by FLUXNET and AmeriFlux.

The third category includes data sources employed for PFTs classification, incorporating 7 site-related articles for C3/C4 classification, flux tower site measurements of precipitation and air temperature, global maps of the Köppen-Geiger climate classification, and the reprocessed MODIS Version 6.1 Leaf Area Index dataset. The Köppen-Geiger climate classification maps, presented at 1 km resolution, are derived from an ensemble of four high-resolution, topographically corrected climatic maps. They demonstrate higher classification accuracy and substantially more detail than previous versions. The reprocessed MODIS LAI used the modified temporal spatial filter (mTSF) method for simple data assimilation, then applied the post processing-TIMESAT (a software package to analyze time-series of satellite sensor data) Savitzky–Golay (SG) filter to obtain the result. Site LAI validation shows that the reprocessed MODIS LAI is much smoother and more consistent with adjacent values than the original MODIS LAI, and closer to site observations (Lin et al., 2023; Yuan et al., 2011).

Finally, three global datasets were used to fill in attribute data of sites lacking site-observed FVC, LAI, and soil texture. LAI filling still uses the reprocessed MODIS LAI, whereas the FVC filling employs a global 300 m PFT map, $PFT_{local}$ (Harper et al., 2023). $PFT_{local}$ incorporates a variety of currently available high-resolution satellite data to
quantify the percentage of PFT in each 300 m pixel worldwide. The 300m resolution is well-matched with the regional extent of the flux tower footprint (Chu et al., 2021), providing representative FVC data. Filling of soil texture uses the Global Soil Dataset for Earth System Models (GSDE) (Shangguan et al., 2014). The GSDE harmonizes data collected from various sources and uses a standardized data structure and data processing procedures to derive the final dataset. It has been extensively applied in earth system models (Dai et al., 2019a).

**2.2 Processing Methods**

We undertook three primary steps to establish the final dataset: site and time period selection, attribute collection, and data processing. First, the data selection process involved picking years with a low gap-filled percentage for fluxes (latent and sensible heat) and vapor pressure deficit (VPD), excluding sites subject to external disturbances or unable to undergo EBC checks. Following that, we collected site-observed vegetation, soil, and topography data. Vegetation attributes include FVC,
maximum LAI and mean canopy height. Soil attributes include soil texture, bulk density, organic carbon concentration and depth. Topography attributes include slope and aspect. Additionally, we obtained the reference measurement heights (for emulating the lowest layer of the atmospheric model to which the LSM would be coupled) of wind speed, air temperature and humidity. Then, we filled in FVC, maximum LAI, and soil texture using global datasets. Finally, the FVC was further broken down into different PFTs. Figure 1 presents a flowchart of the processing pipeline, with each step described in detail
below.

**Table 1.** Summary of the data sources to derive the site attribute dataset.

| Data usage | Name | Sources |
|---|---|---|
| For site and time period selection | PLUMBER2 | Ukkola et al., 2022 |
| Attribute data | Site descriptions in literature (113 articles) | Details in Table S1 |
| | Site regional networks (7 websites) | AmeriFlux[a]; AT − Neu websit[a]; ChinaFlux[c]; European Fluxes[d] Global Monitoring Laboratory[e]; OzFlux[f]; Swiss Fluxnet[g] |
| | Fluxnet BADM | https://fluxnet.org/ |
| | AmeriFlux BADM | https://ameriflux.lbl.gov/ |
| PFT information | Site descriptions in literature (7 articles) | Details in Table S1 |
| | Site measurements of precipitation and air temperature | Ukkola et al., 2022 |
| | Köppen-Geiger climate classification maps | Beck et al., 2018 |
| | Reprocessed MODIS Version 6.1 LAI dataset | Lin et al., 2023 |
| Data filling | $PFT_{local}$ PFTs maps | Harper et al., 2023 |
| | Reprocessed MODIS Version 6.1 LAI dataset | Lin et al., 2023 |
| | Global soil dataset for earth system models | Shangguan et al., 2014 |

[a] https://ameriflux.lbl.gov/, [b] http://www.biomet.co.at/, [c] http://www.chinaflux.org/, [d] http://www.europe-fluxdata.eu/,

[e] https://www.gml.noaa.gov/, [f] https://ozflux.org.au/, [g] https://www.swissfluxnet.ethz.ch/.

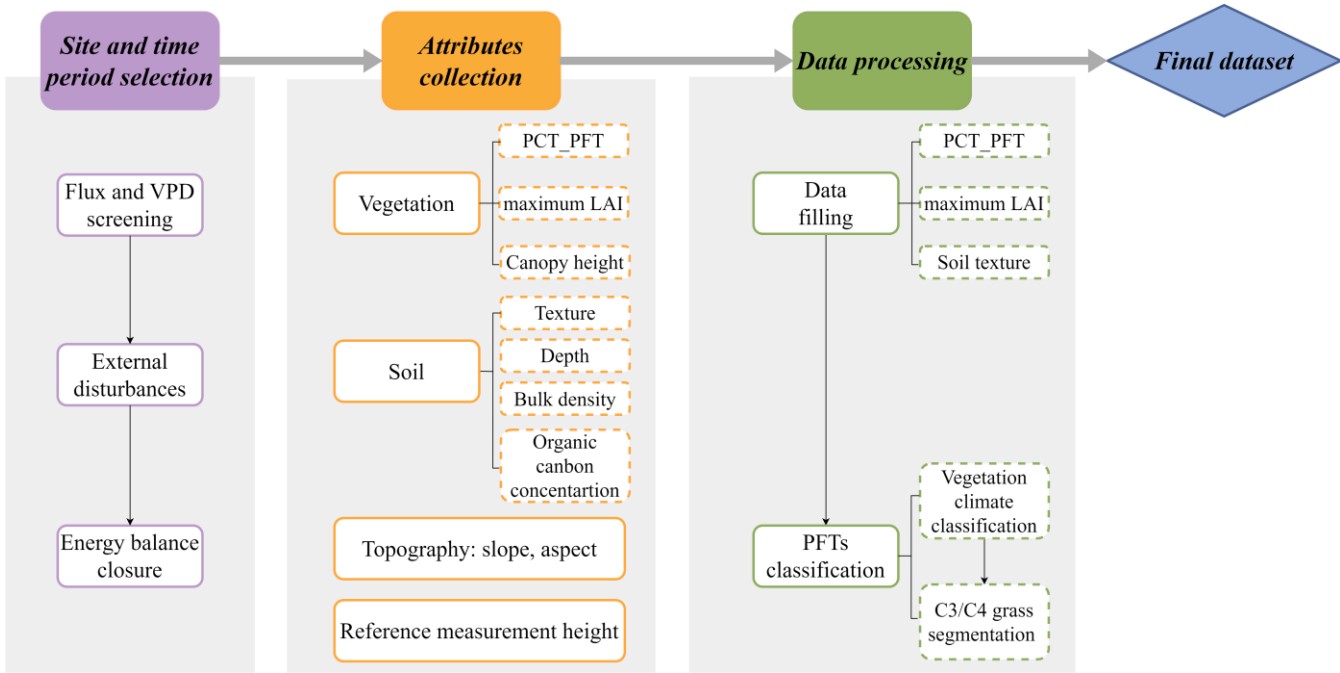

**Figure 1.** Data flow diagram for the generation of the flux tower attribute dataset.

### 2.2.1 Site and time period selection

The PLUMBER2 dataset got 170 sites by screening meteorological data (including five key variables that have the largest influence on LSM simulations: incoming shortwave radiation, precipitation, air temperature, air humidity, and wind speed.). For FLUXNET2015 and La Thuile datasets, specific humidity is not provided in the original data, so it was calculated from VPD (Ukkola et al., 2017). However, the screening process did not consider the gap-filled situation of VPD. As mentioned earlier, it also did not screen the flux variables. To address these limitations, we further implemented quality control on the PLUMBER2 dataset by performing the following three steps:

1. Sites with only one year of observations were excluded to ensure data stability and reliability.
2. Selected the years where the proportion of data with fluxes (latent and sensible heat) quality control (QC) $\leq 1$ exceeds 90 % (QC = 0 denotes observed data, QC = 1 represents high-quality gap-filled data in FLUXNET2015 and La Thuile, no QC = 1 in OzFlux).
3. Selected the years where the proportion of VPD QC = 0 exceeds 90 % in FLUXNET2015 and La Thuile datasets.

Furthermore, we excluded 23 sites that lacked ground heat flux observations because the EBC correction factor ($f_{EBC}$) could not be calculated ($f_{EBC}$ = (Rn - G) / (Qle+ Qh), net radiation (Rn), ground heat flux (G), latent heat flux (Qle)

and sensible heat flux (Qh)). Additionally, two sites (FR-Lq1 and FR-Lq2) were removed as they have a very low energy closure ratio (EBR, calculated as (Qle+ Qh / (Rn – G) according to Wilson et al. (2002)) after performing energy closure (details in Table S3). Lastly, we excluded 10 sites that experienced external disturbances during the observation period, such as irrigation, deforestation, and one site impacted by a large water body nearby (details in Table S3). In the end, we preserved non-consecutive years that met our criteria. This allows us to maximize the utility of valuable observational data. Details of the selected and excluded sites and years are displayed in Tables S2 and S3.

### 2.2.2 Data collection for vegetation attributes

### Percent cover of plant functional types

FVC data was sourced from site descriptions in literature, regional networks, and FLUXNET BADM files. We sought appropriate representations of site FVC and obtained site-observed FVC data for 53 sites. To maximize the amount of FVC collected, some assumptions were made at certain sites during the data collection process, addressing scenarios as follows:

1. For sites lacking explicit FVC data but providing the percentage of vegetation flux footprint contribution or dense forest canopy basal area, we treated these values as FVC. Since FVC directly determines these metrics, and they are numerically similar.

2. In grassland and cropland sites, the vegetation cover type typically exhibits a high degree of homogeneity. Therefore, we referred to site pictures (photographs taken at the site) to make a judgment. If a homogeneous cover could be determined from the pictures, it was assigned a 100 % coverage percentage.

3. Some grassland sites with annual vegetation may experience seasonal bare soil exposure. For these sites, we used the FVC during the peak vegetation growth period.

4. In forest sites, we simply treated forest litter as grass cover in the absence of additional information.

After that, trees and shrubs were classified as evergreen or deciduous, coniferous or broadleaf, based on their vegetation type. As an example, eucalyptus trees are classified as evergreen broadleaf trees. For data completeness, we used the $PFT_{local}$ maps to fill in data for sites lacking site-observed FVC values.

We further broke down the FVC into PFTs to meet the requirements of LSM simulations using PFTs. The breakdown method is as follows: First, the climate type of PFT was determined according to the Köppen climate classification (Poulter et al., 2011). Then, C3 and C4 grasses were partitioned using site descriptions. If site descriptions were unavailable, flux tower air temperature, precipitation, and the reprocessed MODIS LAI are used to calculate LAI proportions under C3/C4 climatic conditions, thereby estimating the C3/C4 grass proportions (Still et al., 2003).

A total of 16 PFTs includes the original set of 15 PFTs initially developed by Bonan et al. (2002) supplemented with a new bare soil surface type. The full set of PFTs includes bare soil; Needleleaf evergreen tree, temperate (ENT_Te); Needleleaf evergreen tree, boreal (ENT_Bo); Needleleaf deciduous tree (DNT); Broadleaf evergreen tree, tropical (EBT_Tr); Broadleaf evergreen tree, temperate (EBT_Te); Broadleaf deciduous tree, tropical (DBT_Tr); Broadleaf deciduous tree, temperate (DBT_Te); Broadleaf deciduous tree, boreal (DBT_Bo); Broadleaf evergreen shrub, temperate (EBS_Te);

Broadleaf deciduous shrub, temperate (DBS_Te); Broadleaf deciduous shrub, boreal (DBS_Bo); C3 grass, arctic; C3 grass; C4 grass; Crop. This PFTs classification scheme is widely utilized in LSMs.

**Maximum leaf area index**

Maximum LAI data were primarily sourced from site descriptions in literature and AmeriFlux BADM files which could be the explicitly stated maximum LAI values or those derived from interannual scatterplots. To maximize data availability, we made the following assumptions at certain sites. Specifically, summertime LAI observation was considered as the maximum LAI. And when a single LAI value was provided without observation time or supporting information, it was accepted as the maximum LAI. To ensure data transparency, quality control flags were implemented in the final dataset, allowing users to select data based on their acceptance criteria. A total of 67 site observations of maximum LAI were collected, with 33 sites providing the year of observation. For data completeness, we used the reprocessed MODIS Version 6.1 LAI dataset to fill in missing site-observed maximum LAI data.

**Canopy height**

We calculated the mean canopy height over the observation period for 69 sites included in FLUXNET2015 dataset, using the canopy heights reported in FLUXNET BADM file across different periods. The mean canopy height provides a more truthful representation of the vegetation condition during the period of observation. For the remaining 21 sites, the canopy height provided by PLUMBER2 was used.

**2.2.3 Data collection for soil attributes**

**Soil texture**

Soil texture data were sourced from site descriptions in literature, regional networks, and AmeriFlux BADM files. These descriptions provided information in two forms: (1) percentages of sand, silt, and clay, and (2) soil texture types, such as sandy loam. For the latter, which do not provide the percentages of sand, silt, and clay, we referred to the soil composition table presented by Dy and Fung (2016) to derive the specific proportions. This table classifies soil into 16 categories based on the proportions of sand, silt, and clay. Overall, 72 site observations of soil texture were collected, with 34 sites providing information on the depth of observations. For data completeness, we used the GSDE dataset to fill in the data for sites lacking site-observed soil texture.

**Soil bulk density, organic carbon concentration, and depth**

Soil bulk density, organic carbon concentration, and depth data were sourced from site descriptions in literature, regional networks, and AmeriFlux BADM file. Specifically, soil bulk density was collected at 37 sites, soil organic carbon concentration at 23 sites, and soil depth at 31 sites. The observation depth was recorded for soil bulk density at 32 sites and for organic carbon concentration at 22 sites. Despite the limited availability of site-observed data for the three soil attributes, we included them in the final dataset. For researchers conducting site-specific studies, these data can serve as valuable references.

**2.2.4 Data collection for topography attributes**

The topography data encompasses site slope and aspect. These data were gathered from site descriptions in literature, regional networks, FLUXNET and AmeriFlux BADM files. Specifically, we acquired slope for 57 sites, and aspect for 49 sites from these sources.

**2.2.5 Reference measurement height**

Site descriptions in literature, regional networks, FLUXNET and AmeriFlux BADM files were all sources for the reference measurement heights. From these sources, we searched for the heights of wind speed, air temperature, and air humidity measurements or the height of the instrument used for these measurements (e.g., wind cups and temperature and humidity sensors). In cases where the flux tower meteorological observation equipment lacked a dedicated wind speed measurement device, we assumed that the use of a three-dimensional sonic anemometer for wind speed measurements. Consequently, wind observation heights were available for a total of 76 sites, while 65 sites had temperature and humidity observation heights. For the remaining sites where observation heights were not reported, we used the flux observation height as a substitute.

**2.3 Modeling assessment of attribute data**

The impact of collected attributes on carbon, water, and energy fluxes is assessed through single-point simulations using the latest version of the Common Land Model (Dai et al., 2003) (CoLM202X, https://github.com/CoLM-SYSU/CoLM202X/tree/master, last access: 21 November 2023). CoLM202X incorporates processes related to biogeophysics, biogeochemistry, ecological dynamics and human activities, and offers optional processes and schemes which can be customized by the user. In our experiments, vegetation is modeled using a set of time-invariant parameters (optical properties: leaf optical properties; morphological properties: canopy height, vegetation root depth and profile, leaf size and angle distributions; and physiological properties). The dynamic vegetation module is turned off and the time-variant LAI and stem area index (SAI) values are prescribed from the reprocessed MODIS LAI data (Lin et al., 2023; Yuan et al., 2011). The two-big-leaf model (Dai et al., 2004) is employed to calculate processes such as radiative transfer (Yuan et al., 2017), photosynthesis (Collatz et al., 1992; Farquhar et al., 1980), and stomatal conductance (Ball et al., 1987). Surface turbulent exchange is simulated using similarity theory (Brutsaert, 1982; Zeng and Dickinson, 1998). Total evapotranspiration includes evaporation from stems, leaves, and the ground, as well as vegetation transpiration. Surface and subsurface runoff consider factors such as terrain, groundwater level, precipitation, and infiltration rate. Additionally, the model accounts for processes including precipitation phase and intensity, canopy interception, vertical movement of water in snow and soil, and snow compaction (Dai et al., 2003).

The simulations aim to evaluate the differences in model results between runs using site-observed attributes and those commonly utilized by LSMs. For simplicity, we refer to site-observed data as "site data" and data commonly utilized by

LSMs as "default data" in subsequent descriptions. We focus on four crucial attributes, PCT_PFT, LAI, canopy height and soil texture, to demonstrate their corresponding impacts. In site data simulations, we scaled the default LAI time series to match the maximum LAI observed, corrected the default canopy height using site canopy height, and replaced the default topsoil texture (0-28.9 cm) with the site-observed soil texture. For sites with multiple PFTs, we calculated the LAI for each PFT using growing degree days and PCT_PFT values (Lawrence and Chase, 2007). Canopy height was divided into three categories based on PFTs (trees, shrubs, or grassland), using site data to adjust the default values for the corresponding group, while the other two groups retained their default values.

The default data generally rely on global LAI and soil texture mapping products, lookup table canopy height, and site IGBP (International Geosphere–Biosphere Programme) classifications to characterize surface vegetation and soil conditions. In this study, the default LAI and soil texture refer to the reprocessed MODIS version 6.1 LAI and the GSDE soil texture shown in Table 1. Lookup table canopy heights are sourced from CoLM, while site IGBP classifications are obtained from FLUXNET and OzFlux. We selected ten sites for each attribute—LAI, canopy height, and soil texture—where site data differ most from default data (In the lookup table canopy height simulations, sites with zero plane displacement exceeding reference measurement height are excluded). For the PCT_PFT analyses, sites with IGBP types that include combinations of trees and grasses (OSH, WSA, SAV) were chosen, resulting in six available sites. Table 2 provides an overview of the selected sites along with their corresponding attribute information. Each site was simulated under three conditions: 1) using site data for all attributes at each site, 2) using default data for all attributes at each site, and 3) using default data for the corresponding attribute at sites selected for each attribute separately, while maintaining site data for the remaining attributes. The comparison between simulations (1) and (3) aims to demonstrate the individual impact of each attribute, while the comparison between simulations (1) and (2) shows the combined impact of all four attributes.

At each site, we ran CoLM at either the half-hourly or hourly time resolution, depending on the forcing data provided, for all years in the original dataset. Subsequent analyses were conducted only for the years we selected. To reach an equilibrium in soil moisture and temperature, CoLM loops the atmospheric forcing data for each site's observation period until it reaches 50 years long. The discrepancy between site data and default data is compared by variables related to land surface energy, water, and photosynthesis processes, including latent heat (Qle), sensible heat (Qh), net radiation (Rn), upward shortwave radiation (SWup), gross primary production (GPP), friction velocity (Ustar), surface soil water content (0-4.5cm) (SWC), and total runoff (TR).

To quantify the differences between the output from site data and default data while accounting for seasonal fluctuations in the impacts of soil and vegetation on climate-related variables (Dirmeyer, 2011; Forzieri et al., 2020), we designed a statistical indicator called the percentage of mean difference (MD %) (Eq. 1). This indicator is calculated by expressing the mean difference for each month as a percentage of the observed or default modeled annual mean. We used multi-year average time series to capture more stable differences in output. In addition, we used delta root mean squared error (ΔRMSE) (Eq. 3) and Δ|Bias| (Eq. 5) to measure the differences in RMSE and Bias of the output between site and default data, allowing us to assess the model's performance after incorporating site data.

$$MD \% = \begin{cases} \dfrac{|\frac{1}{n}\sum_{i=1}^{n}(Mod_{site,i} - Mod_{default,i})|}{\frac{1}{365}\sum_{j=1}^{365}Obs_j}, & for\ Qle, Qh, Rn, SWup, GPP, and\ Ustar \\ \dfrac{|\frac{1}{n}\sum_{i=1}^{n}(Mod_{site,i} - Mod_{default,i})|}{\frac{1}{365}\sum_{j=1}^{365}Mod_{default,j}}, & for\ SWC\ and\ TR \end{cases} \quad n = days\ of\ month \quad (1)$$

$$RMSE = \sqrt{\frac{\sum_{i}^{n}(Mod_i - Obs_i)^2}{n}} \qquad (2)$$

$$\Delta RMSE = RMSE_{site} - RMSE_{default} \qquad (3)$$

$$Bias = \frac{\sum_{i}^{n}((Mod_i - Obs_i)}{n} \qquad (4)$$

$$\Delta|Bias| = |Bias_{site}| - |Bias_{default}| \qquad (5)$$

Where $Mod_{site,i}$ and $Mod_{default,i}$ are the predicted value using site data and default data, respectively. $Obs_j$ is observed value. $n$ is the number of paired values. $RMSE_{site}$ and $RMSE_{default}$ are the RMSE of the simulation results using site data and default data, respectively. $Bias_{site}$ and $Bias_{default}$ also correspond to the Bias in these results.

**Table 2.** Selected sites and their attribute values used in the modeling assessment for attribute data. The suffix "default" denotes default data, and the "site" represents site data.

| Site_LAI | Lat | Lon | LAI_max_default[a](m²/m²) | LAI_max_site[b](m2/m2) |
|---|---|---|---|---|
| DE-Bay | 54.14 | 11.86 | 3.6 | 6.5 |
| DE-Gri | 50.94 | 13.51 | 6.5 (2004[c]) | 4.4 (2004) |
| DK-Lva | 55.68 | 12.08 | 3.1 (2004) | 6.9 (2004) |
| DE-Seh | 58.87 | 6.44 | 3.2 (2009) | 5.9 (2009) |
| IT-Cpz | 41.70 | 12.37 | 5.4 | 3.5 |
| US-GLE | 41.36 | -106.24 | 1.5 | 3.8 |
| US-Goo | 34.25 | -89.87 | 4.5 | 2.0 |
| US-KS2 | 28.60 | -80.67 | 6.6 (2005) | 2.7 (2005) |
| US-MMS | 39.32 | -86.41 | 7.0 | 5.2 |
| US-MOz | 38.74 | -92.20 | 6.1 (2006) | 4.0 (2006) |
| **Site_TEX** | **Lat** | **Lon** | **TEX-default[d]** | **TEX_site[b]** |
| AU-Cpr | -34.00 | 140.58 | 64/18/18 | 94/4/2 |
| AU-DaP | -14.06 | 131.31 | 63/18/19 | 92/5/3 |
| AU-DaS | -14.15 | 131.38 | 63/12/25 | 92/5/3 |
| CZ-wet | 49.02 | 14.77 | 39/37/32 | 10/85/5 |
| DE-Gri | 50.94 | 13.51 | 52/29/20 | 10/81/9 (0-23cm) |
| ES-LMa | 39.94 | -5.77 | 49/24/24 | 80/11/9 (0-30cm) |
| FI-Sod | 67.36 | 26.63 | 52/25/20 | 92/5/3 |
| IT-Cpz | 41.70 | 12.37 | 33/45/22 | 87/8/5 (0-10cm) |
| IT-SRo | 43.72 | 10.28 | 69/17/15 | 95/4/1 (10-20cm) |
| SD-Dem | 13.28 | 30.47 | 67/18/14 | 96/4/0 |

| Site_HTOP | Lat | Lon | $H_{can}$_default[e] (m) | $H_{can}$_site[b] (m) |
|---|---|---|---|---|
| AU-Lit | -13.17 | 130.79 | 35 | 20.0 |
| BE-Vie | 50.30 | 5.99 | 17 | 33.7 |
| CH-Dav | 46.81 | 9.85 | 17 | 25 |
| DE-Hai | 51.07 | 10.45 | 20 | 33.9 |
| DE-Tha | 50.93 | 13.56 | 17 | 28.4 |
| IT-Cpz | 41.70 | 12.37 | 35 | 14.3 |
| IT-Lav | 45.95 | 11.28 | 17 | 28.0 |
| IT-Ren | 46.58 | 11.43 | 17 | 29.0 |
| RU-Fyo | 56.46 | 32.92 | 17 | 26.3 |
| US-Ton | 38.43 | -120.96 | 20 | 9.9 |

| Site_FVC | Lat | Lon | IGBP | PCT_PFT_site[b] |
|---|---|---|---|---|
| AU-How | -12.49 | 131.14 | WSA | EBT_Tr/DBS_Te/C4 : 50/25/25 |
| ES-LMa | 39.94 | -5.77 | SAV | EBT_Te/C3 : 20/80 |
| SD-Dem | 13.28 | 30.47 | SAV | EBT_Tr/C3/C4 : 10/27/63 |
| US-SRM | 31.82 | -110.86 | WSA | DBS_Te/C3/C4 : 35/43/22 |
| US-Ton | 38.43 | -120.96 | WSA | EBT_Te/C3 : 40/60 |
| US-Whs | 31.74 | -110.05 | OSH | Bare/DBS_Te/C3 : 39/51/10 |

[a] The maximum LAI at the pixel containing the site provided by Reprocessed MODIS version 6.1 LAI. [b] Site-observed data collected in this study. [c] Specific year of maximum LAI. [d] The top layer soil texture (sand/silt/clay) at the site location extracted from the GSDE dataset. [e] Canopy height of the dominant vegetation type at the site from the CoLM lookup table.

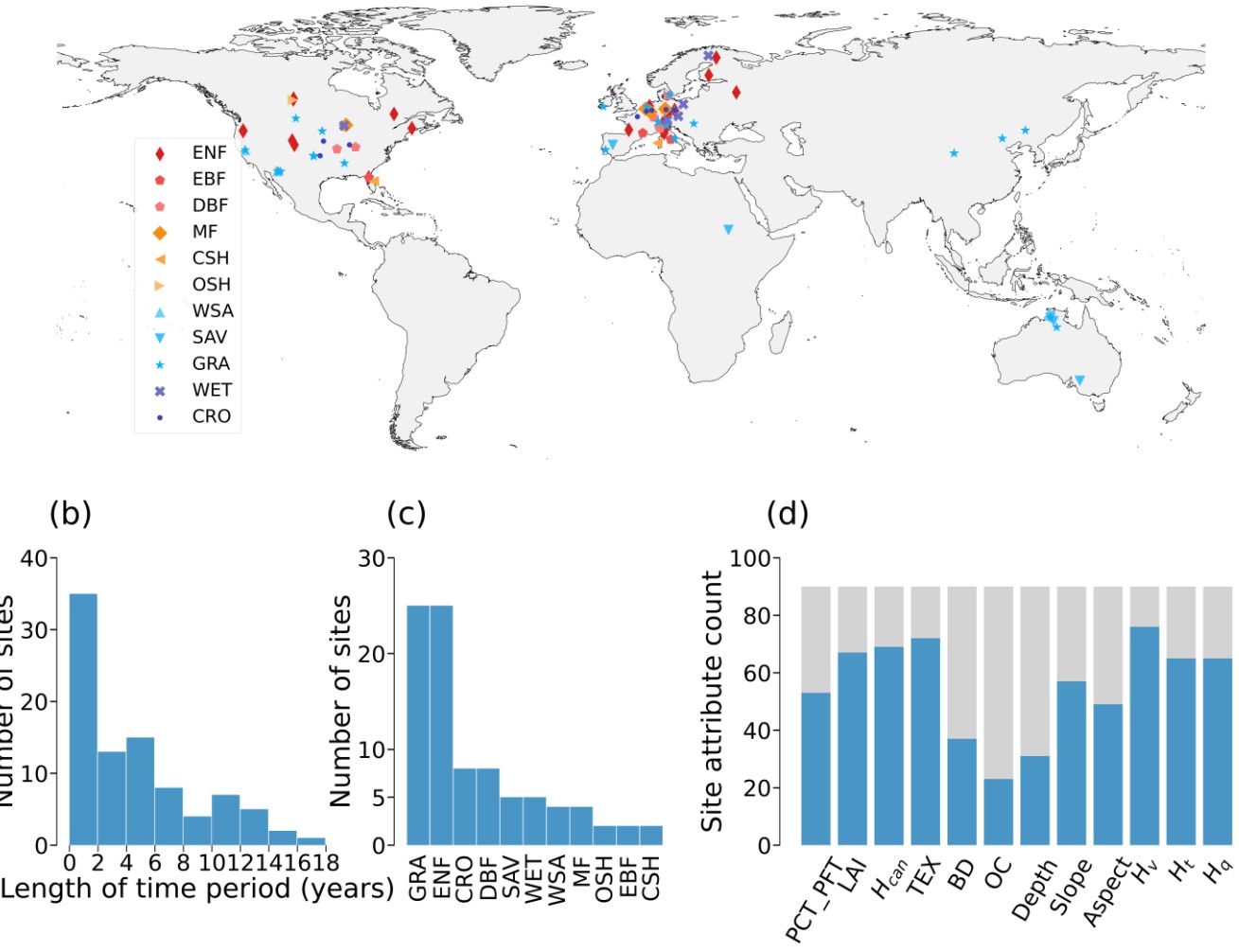

**Figure 2.** Summary of selected sites and collected site-observed attributes. (a) Geographical distribution of selected sites and their IGBP types. (b) A histogram showing the number of sites based on the number of years of selected sites. (c) Number of selected sites per IGBP vegetation class. (d) Number of collected site-observed attributes for percent cover of PFTs (PCT_PFT), maximum LAI (LAI), mean canopy height ($H_{can}$), soil texture (TEX), bulk density (BD), organic carbon concentration (OC), and soil depth (Depth), slope, aspect, and reference measurement heights (Wind speed: $H_v$; Air temperature: $H_t$; Humidity: $H_q$).

## 3 Results

### 3.1 Global distribution and attribute information of selected sites

The final dataset contains 90 globally distributed sites (Fig. 2a). The majority are in North America and Europe, followed by Australia, with smaller representations in Asia (3 sites) and Africa (1 site). Temporal coverage spans from 1997 to 2017, totaling 475 site years. Individual site observations range from 1 to 17 years, with a median of 4 years (Fig. 2b). Despite a reduction in available sites and years due to rigorous quality control, the dataset does offer reliable meteorological forcing and flux assessment data for LSMs. Furthermore, the 90 sites encompass the full range of IGBP classifications originally

presented, covering a wide spread of biomes, from grasslands and savannas to forest ecosystems (Fig. 2c). This enables users to evaluate models across diverse biomes using quality-benchmarked flux tower observations.

Out of the 90 sites, data were collected on PCT_PFT for 53 sites, maximum LAI for 67 sites, average canopy height for 69 sites, and soil texture for 72 sites. Additionally, soil bulk density was available for 37 sites, soil organic carbon concentration for 23 sites, and soil depth for 31 sites. Data on slope were collected for 57 sites, aspect for 49 sites, wind

observation height for 76 sites, and air temperature and humidity observation heights for 65 sites (Fig. 2d). In the absence of site-observed PCT_PFT, soil texture, and LAI, we opted for appropriate global data to fill in these missing for data completeness. To improve data utilization, we provide the observation year of maximum LAI and the depth of soil texture, which are available at 33 and 34 sites, respectively.

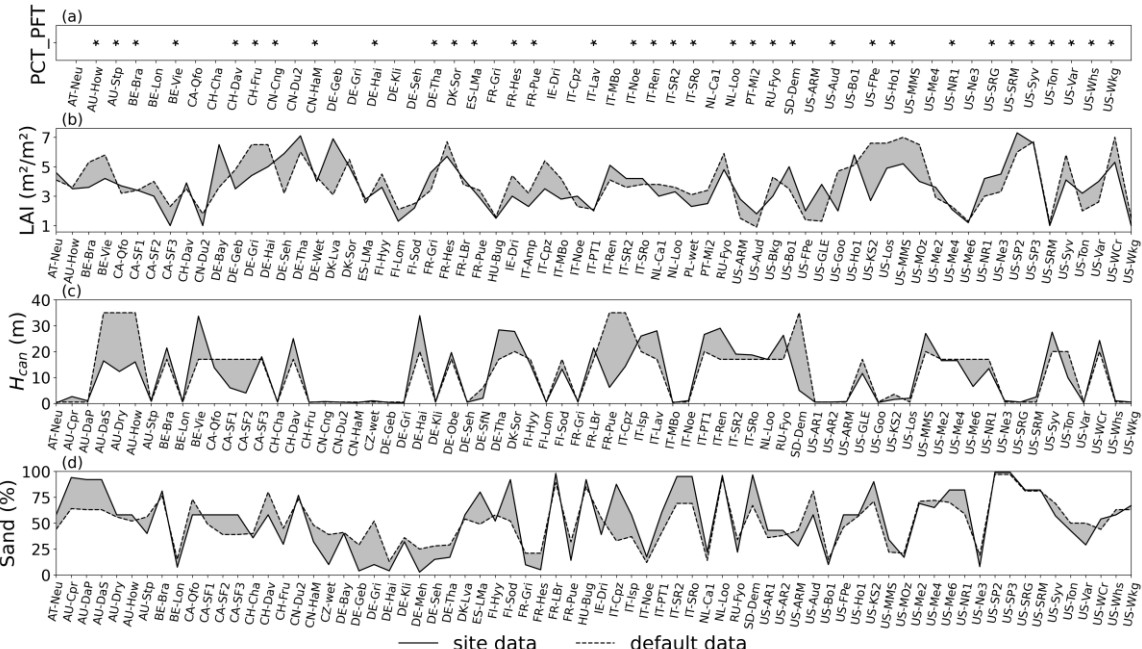

**Figure 3.** The discrepancies between site data and default data of (a) percent cover of PFTs (PCT_PFT), the asterisk

indicates non-single PFTs, (b) maximum LAI, (c) canopy height ($H_{can}$), and (d) the percentage of sand.

Figure 3 depicts the discrepancies between site data and default data for PCT_PFT, maximum LAI, canopy height, and soil texture. The PCT_PFT shows multiple PFTs at 34 sites, offering a more accurate representation of vegetation conditions compared to IGBP classifications. For LAI, canopy height, and soil texture, variations between site data and default data are substantial at certain sites. Specifically, at 31 sites, discrepancies in LAI values exceed 1 m²/m²; canopy height differs by over 10 meters at 15 sites, and sand percentage varies by more than 20% at 18 sites.

## 3.2 The flux tower site attribute dataset

The final dataset is formatted in NetCDF (Network Common Data Form). Table 3 outlines the attribute variables and corresponding descriptions for each site in the file. These attributes can be categorized into vegetation, soil, and topography attributes, as well as reference heights and filtered high-quality years.

For maximum LAI, the file provides both the year range covered by maximum LAI and the maximum value for a specific year. Regarding the three soil attributes, soil texture, bulk density, and organic carbon concentration, the file provides values for multiple soil layers, along with the specific depth of each layer. Concerning reference height, we give its corresponding observed variable, i.e., wind speed, air temperature and humidity, or fluxes (latent and sensible heat). Additionally, the NetCDF file incorporates reference sources for each attribute. These sources are included to facilitate access to the original data and enhance flexibility in application. A summary of these reference sources is presented in Table S1.

**Table 3.** Attribute variables and their descriptions included in the final dataset (note that not all sites provide 'Soil_BD', 'Soil_OC', 'Soil_depth', 'Slope', and 'Aspect').

| Variable (Dimension) | Long name | Unit | Description |
|---|---|---|---|
| PCT_PFT (pft=16) | Percent plant functional types cover | % | Source[a]; |
| LAI_Max | Maximum leaf area index | $m^2/m^2$ | Source; year_range[b]; LAI_Max_year[c] |
| Canopy_height | Canopy height | m | Source; |
| Soil_TEX (particle_size=3, soil_layer=4) | Soil texture(sand/silt/clay) | % | Source; layer_n_depth[d] |
| Soil_BD (soil_layer=4) | Soil bulk density | $g\ cm^{-3}$ | Source; layer_n_depth[d] |
| Soil_OC (soil_layer=4) | Soil organic carbon concentration | % | Source; layer_n_depth[d] |
| Soil_depth | Soil depth | cm | Source |
| Slope | Site slope | - | Source; |
| Aspect | Site aspect | - | Source; |
| Reference_height_v | Measurement height of wind speed or flux | m | Source; Measurement variable (Wind or Flux) |
| Reference_height_t | Measurement height of air temperature or flux | m | Source; Measurement variable (Temperature or Flux) |
| Reference_height_q | Measurement height of air humidity or flux | m | Source; Measurement variable (Humidity or Flux) |
| year_qc (year=21) | Selected year of high-quality data | - | - |

[a] The sources of collected attribute data. [b] The year range covered by maximum LAI. [c] Maximum LAI for specific year. [d] The "n" ranges from 1 to 4, denoting the four soil layers in ascending order of depth. The parameter "layer_n_depth" indicates the depth of respective "soil layer", corresponding to the depth at which soil data is observed.

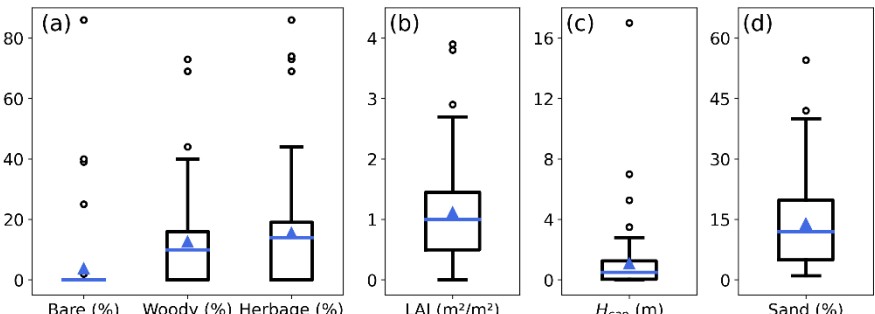

**Figure 4.** Quantification of discrepancies between site data and filled data for (a) PCT_PFT, (b) maximum LAI, (c) canopy height, and (d) percentage of sand (at all sites for which both types of data are available). The 16 PFTs were divided into three main categories (bare soil, woody, and herbaceous vegetation) for separate quantification. Boxes (25th and 75th percentiles) and whiskers (5th and 95th percentiles), with median (blue line) and mean (blue triangle). Hollow circles denote outliers defined as values greater than 1.5 times the interquartile range from the nearest 25th or 75th percentile.


     Figure 4 quantifies the differences between site data and filled data for sites where both data sources are available, illustrating the inhomogeneities in the final dataset resulting from data filling. Differences in vegetation cover (including bare soil, woody, and herbaceous vegetation) generally fall within 20 %, with a minority of sites exceeding 40%. The mean and median LAI differences are approximately 1 m²/m². Canopy height deviations are primarily within 2 m, although a few
sites exceed 4 m. Differences in sand content typically remain within 30 %, with both mean and median differences below 15 %. This quantification suggests that the filled data are generally reliable across most sites.

### 3.3 Impact of site attributes on modeling

     The impacts of altering land surface representation from default data to site data, quantified by MD %, on Qle, Qh, Rn, SWup, GPP, Ustar, SWC, and TR are shown in Fig. 5. It distinctly demonstrates how vegetation and soil components affect
carbon, water, and energy fluxes to varying degrees, contingent on the season. The impacts of vegetation cover, soil texture, and LAI on Qle and Qh is primarily observed in the spring and summer, while canopy height exerts its most substantial effects in autumn and winter. The impact of vegetation cover on Rn and SWup remains consistent throughout the year, whereas LAI maintains a more pronounced effect in spring and summer. In terms of GPP, attributes play a more significant role during the summertime. However, the effects of vegetation and soil attributes on Ustar appear to be independent of
season. SWC and TR are both predominantly influenced by soil texture. The difference is that soil texture significantly

affects SWC across all seasons, whereas its impact on TR occurs primarily during the summer and fall. Additionally, vegetation cover was observed to have a significant effect on TR at the SD-Dem site. This is due to the salient impact at the SD-Dem site, situated within the African savannah with an average annual precipitation of 320 mm (Ardö et al., 2008).

To elucidate the magnitude of each attribute's impact on different variables, figure 6 further displays the monthly average maximum MD %. On average, changes in latent and sensible heat are not dominated by any single attribute. All four attributes—PCT_PFT, LAI, canopy height, and soil texture—have a relatively strong impact on both. Their monthly average maximum MD % on Qh is all in the range of 14-36 %. And the effect of soil texture on Qle is comparatively greater, at 18.3 %. Regarding Rn, vegetation cover emerges as the chief influencer with a monthly average maximum MD % of 8.8 %. In contrast, SWup is heavily dictated by LAI, at 56.7 %, due to the exceptionally high value at the US-GLE site. Vegetation cover and LAI, both with a monthly average maximum MD % of more than 50 %, dominate the changes in GPP. Soil texture also has a visible impact on GPP, due to its influence on soil permeability, aeration, and the capacity to retain water and nutrients. On the other hand, Ustar is almost exclusively shaped by vegetation cover and canopy height. This makes sense because the intensity of land-atmosphere exchange in vegetated areas is directly tied to canopy height, and changes in vegetation cover typically correspond to changes in canopy height. Concerning SWC and TR, vegetation cover and soil texture are the two crucial attributes. Soil texture exhibits monthly average maximum MD % of 46.3 % for SWC and 129.8 % for TR, while vegetation cover shows 22.7 % and 293.8 %, respectively.

Figure 7 uses ΔRMSE and Δ|Bias| to show the shifts in model performance using site data. The incorporation of site-observed attribute data significantly improves the simulation of Rn, SWup, and Ustar. Concerning individual attributes, PCT_PFT proves particularly beneficial for modeling both Rn and SWup. Concurrently, including site LAI also enhances the simulation of SWup. Improvements in these fundamental energy terms contribute to more accurate modeling of latent and sensible heat. Furthermore, site LAI and canopy height demonstrates steady improvements on GPP and Ustar, respectively.

In summary, these results underscore the significant impact and importance of incorporating site-observed attribute data in the simulation of carbon, water, and energy fluxes in LSMs.

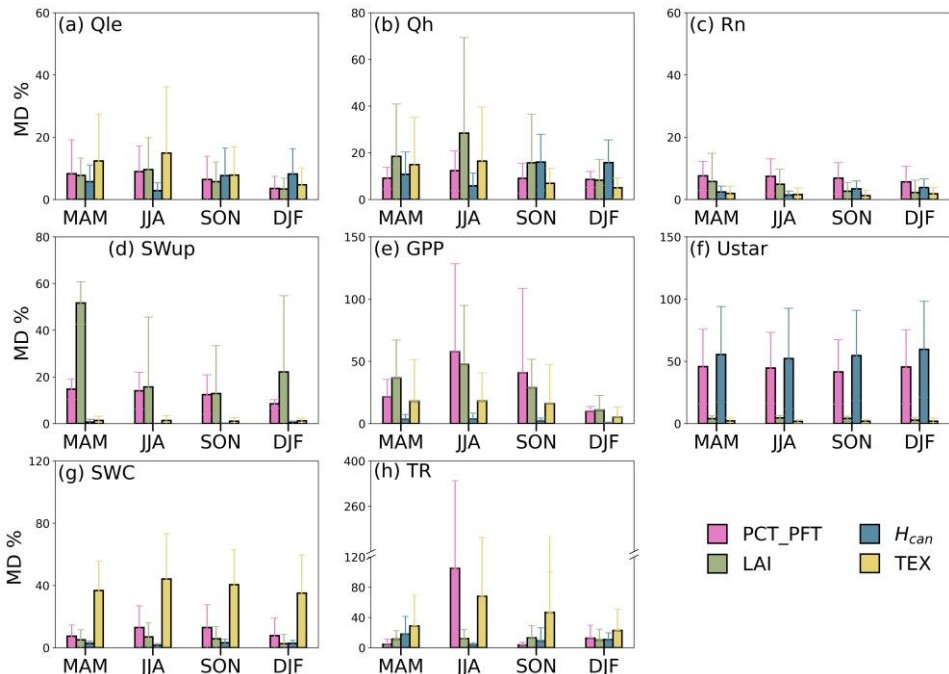

**Figure 5.** Percentage of mean difference (MD %) of PCT_PFT, LAI, canopy height ($H_{can}$) and soil texture (TEX) on Qle, Qh, Rn, SWup, GPP, Ustar, SWC and TR for each season, respectively. Error bars indicate one standard deviation from the multi-site mean. Monthly adjustments were made for Southern Hemisphere sites, to ensure consistency between seasons and months in multi-site averaging (i.e., DJF is considered JJA, MAM is considered SON, and vice versa).

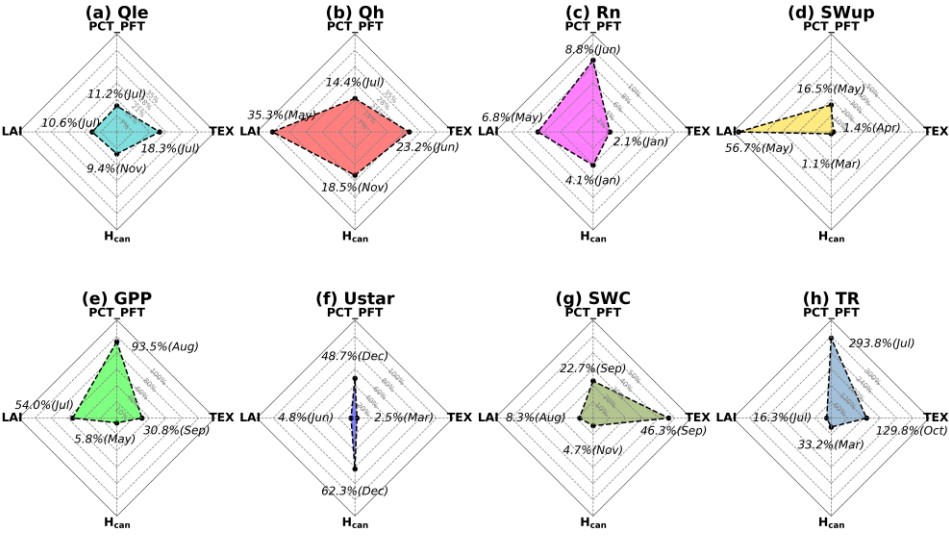

**Figure 6.** Monthly average maximum MD % of PCT_PFT, LAI, canopy height ($H_{can}$) and soil texture (TEX) on Qle, Qh, Rn, SWup, GPP, Ustar, SWC and TR, respectively. The month of occurrence for each maximum value is indicated in parentheses.

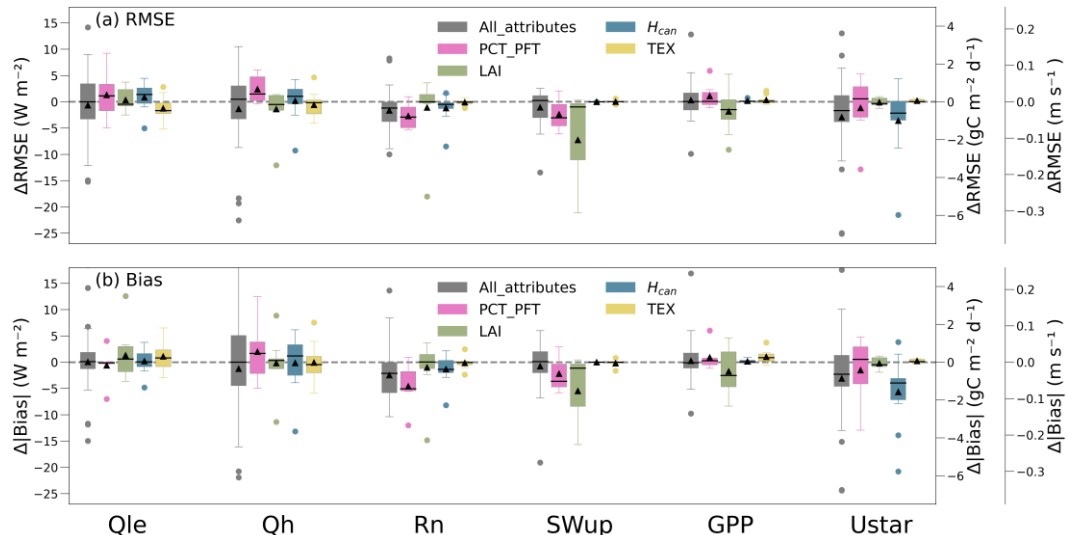

**Figure 7.** Box plot of changes in RMSE (ΔRMSE) and absolute Bias (Δ|Bias|) when using site data versus default data. PCT_PFT, LAI, $H_{can}$ (canopy height), and TEX (soil texture) denote the individual impacts of the four attributes. All_attributes represents the changes produced by four attributes together across the 36 sites selected. Boxes (25th and 75th percentiles) and whiskers (5th and 95th percentiles), with median (black line) and mean (black triangle). Solid circles denote outliers defined as values greater than 1.5 times the interquartile range from the nearest 25th or 75th percentile.

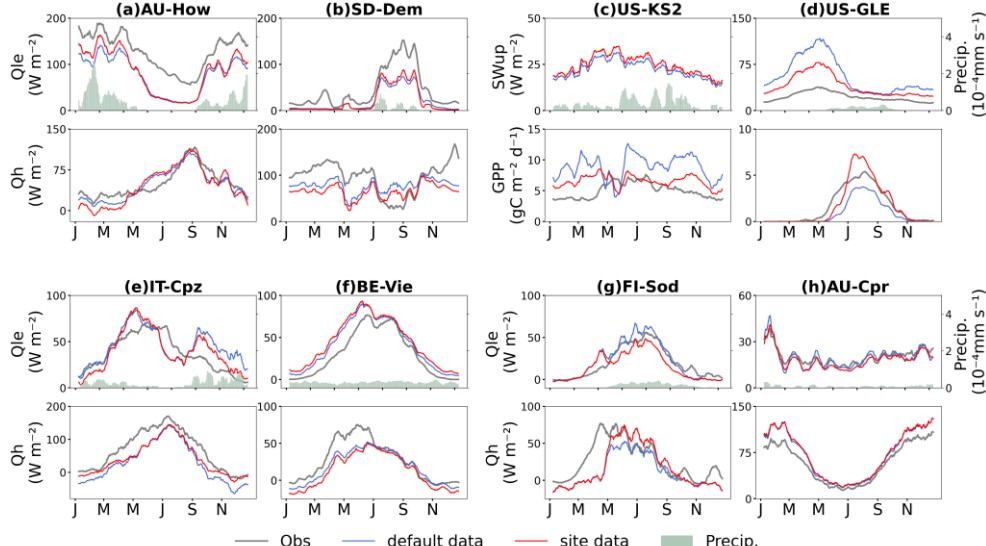

**Figure 8.** Multi-year daily average of the seasonal cycle of model (default, site) and observed Qle, Qh, SWup, and GPP at 8 selected sites. Two sites were chosen for each attribute for comparison: PCT_PFT (AU-How and SD-Dem), LAI (US-KS2 and US-GLE), $H_{can}$ (IT-Cpz and BE-Vie), Soil texture (FI-Sod and AU-Cpr). Data are smoothed with a 14-day moving average for clarity.

## 4 Discussion

In land surface community, flux tower attribute data currently does not receive sufficient attention. However, the site attribute data are nearly as critical as the flux tower observations themselves. We hope that future flux tower datasets will provide standardized site attributes. In this study, we have acquired 90 sites with high quality by a comprehensive selection process, and providing extensive site-observed data on vegetation, soil, and topography attributes. Through single-point simulations, we demonstrated their indispensable role in LSM development. Accurate attribute data will provide multiple benefits by lowering uncertainty in model calibration and evaluation.

After selection, fewer sites and years are available. However, the retained data offers trustworthy observations that can be directly applied. Data quality is generally the focus of model calibration and evaluation, and developing LSMs can benefit immensely from using a modest number of sites (Brooke et al., 2019; Harper et al., 2021; Swenson et al., 2019). Therefore, these updates will help the model's developments. To collect more site-observed attribute data, while considering the diversity described within the same attribute data, particularly the percentage of vegetation cover, we made a few approximations and assumptions during data collection procedure, such as using approximation substitution and site photographs to assist in judgment. Although these methods may introduce slight deviations, they do a good job of reproducing the surface conditions of these sites. Furthermore, we provide descriptions of the attribute data as detailed as possible. For instance, the year and depth of observation are given along with the maximum LAI and soil texture whenever feasible, respectively. They are valuable references for data applications. One might argue that the auxiliary descriptions are just as important as the attribute data itself.

Using CoLM at 36 sites, we evaluated the impacts of PCT_PFT, LAI, canopy height, and soil texture on model results. What is conducted here is not an ideal experiment, but rather an actual demonstration of the discrepancies in model results between site data and default data. The results are in line with previous research (Dai et al., 2019b), showing that vegetation cover appreciably affects each of the eight variables examined, often being the dominant attribute (Fig. 6). This is due to plant cover being the most prominent surface feature, directly altering surface energy absorption. The net radiation simulation was improved using the site PCT_PFT, but the performance of latent and sensible heat was not as good. This may be related to uncertainties in the model itself as well as other input data. Such as vegetation biophysical parameters, soil thermal and hydraulic conductivities, etc.

Additionally, we find that the impact of attributes is substantially associated with precipitation. As illustrated in the average seasonal cycle shown in Fig. 8. At the AU-How site in Australia, ample rainfall during the wet season, combined with the increase in surface available energy due to vegetation cover, brings about a significant increase in Qle. In contrast, since limited water is available for evapotranspiration at the SD-Dem site, Qh is the primary feedback from changes in surface energy. The results from the US-KS2 and US-GLE sites indicate that the growing season, synchronized with water availability, is when LAI exerts a major influence on GPP. Furthermore, a notable variation in SWup was seen at the US-GLE site, attributed to the presence of snow cover (Berryman et al., 2018). Corrections to LAI can improve the simulation

by reducing albedo inaccuracies. This corroborates Essery (2013) point that inadequate land-cover data is largely to blame for the uncertainty in the climate–snow albedo feedback in LSMs. Results from the IT-Cpz and BE-Vie sites suggest that

differences in the intensity of land-air exchange, caused by variations in canopy height, are clearly reflected in Qle during the rainy season. Regarding soil texture, a comparison between FI-Sod and AU-Cpr sites revealed stronger control of Qle by soil texture during the period of high precipitation intensity. This is partly attributed to increased water availability and largely to the pronounced differences in soil infiltration capacity under high-intensity precipitation events.

A previous study by Ménard et al. (2015) stated that attribute data have little effect on modeling results. This study,

however, may lack representativeness since it was limited to one site. Furthermore, it averaged differences resulting from attribute data across the whole time series by using the raw RMSE and correlation coefficient statistical metrics. This approach makes it difficult to detect the crucial role of attribute data. As described in Sect. 3.3, the impacts of attribute data on climate-related variables generally occur over specific periods (mostly during the growing season) rather than throughout the year.

By combining multiple data sources, we were able to maximize the available site-observed attribute data. Nevertheless, the data sources were primarily from published works, which led to some missing data at certain sites. The attribute data focused only on soil and vegetation information. Future endeavors should incorporate additional surface parameters, such as irrigation, wildfire, and the depth of soil moisture and vegetation roots, which are required for LSMs. Such observations and collections of site time-invariant attributes are generally low-cost but would strongly benefit model enhancement. In addition,

the impact of attribute data on model results was assessed using one model, potentially limiting the representativeness of our findings.

As LSMs continually advance their schemes and processes, an increasing array of surface parameters will be incorporated, elevating the models to a heightened level of complexity. It is imperative that these parameters be clearly defined and prescribed. Working with site-observed attribute data enabled us to narrow down reasons for model biases,

thereby enhancing our understanding of the true effects of diverse schemes and processes.

## 5 Data availability

The flux tower site attribute dataset provides comprehensive filtered high-quality years, site-observed vegetation, soil, topography attributes, and reference measurement heights. Each site's data is formatted within a NetCDF file named according to the site name, database, and attributes (vegetation, soil, topography, and reference height), such as 'AT-

Neu_FLUXNET2015_Veg_Soil_ Topography_ReferenceHeight.nc'. The dataset comprises a total of 90 NetCDF files and can be accessed at Zenodo under https://doi.org/10.5281/zenodo.12596218 (Shi et. al., 2024).

**6 Code availability**

The processing codes are available at https://github.com/Mbnl1197/Flux-tower-attribute-for-LSM (last access: 4 Sep 2024)
(DOI: https://doi.org/10.5281/zenodo.13684992; Shi et. al., 2024)

**7 Conclusions**

This study is centered on two issues with utilizing flux tower data in LSMs: inadequate data quality and insufficient site attributes. We performed a comprehensive quality control on flux tower data. By examining observation percentage, energy balance closure, and external disturbances, 90 high-quality flux tower sites with 475 site years were produced. By combining
various data sources, we created a flux tower attribute dataset through data collection, processing, and filling procedures. This dataset includes the site-observed PCT_PFT, maximum LAI, mean canopy height, soil properties (texture, bulk density, organic carbon concentrations, and depth), site topography (slope and aspect), as well as the reference measurement heights.

Furthermore, the attribute data collected in this study and frequently used by LSMs are incorporated in single-point modeling respectively, aimed at quantifying the differences in model output. Our results demonstrate the significance of
certain attributes in the variation of specific variables. All four attributes significantly influence both latent and sensible heat. Their monthly average maximum MD % typically ranges from 10 % to 30 %. Vegetation cover and LAI serve as the primary controls for net radiation and upward shortwave radiation, respectively, with monthly average maximum MD % of 8.8 % and 56.7 %. Both GPP and Ustar were strongly influenced by vegetation cover, with LAI and canopy height also exerting significant effects on GPP and Ustar, respectively. The monthly average maximum MD % for each of these impacts exceeds
50 %. For hydrologic variables, i.e., SWC and TR, soil texture typically holds greater significance, followed by vegetation cover. We reveal that the magnitude of these differences is usually accompanied by seasonal fluctuations. Particularly regarding fluxes and GPP, greater discrepancies are generally observed during spring and summer. These results stress the necessity of site-observed attribute data in the development of LSMs.

Our endeavors mitigate the inadequacies of flux tower attribute data, enhancing the ability of flux tower data to serve as
benchmarking data for LSMs. The dataset provides relatively complete site attribute data and high-quality flux validation data, making it suitable for direct use as inputs and for simulation validation in LSMs. This facilitates the comparison of LSM simulations under the same standard framework, promoting their development. Moreover, this effort will draw more attention to flux tower attribute data from the land surface modeling group and foster communication between ecology and modeling communities. We strongly advocate for the routine release of attribute data as part of flux tower data. Making such
ancillary data more easily and routinely accessible would greatly increase the value and usability of the data.

**Author contributions**

Conceptualization: H.Y.; Data curation: H.Y. and J.S.; Formal analysis: J.S., H.Y., W.D. and W.L.; Funding acquisition: H.Y. and Y.D.; Investigation: H.Y. and J.S.; Methodology: H.Y., J.S., N.W., Z.W. and S.L.; Resources: Y.D. and H.Y.;

Software: J.S., H.Y., H.L., W.L., N.W., J.Z. and H.Z.; Validation: H.Y., Z.L., W.D., W.L., S.Z. and X.L.; Visualization: J.S.; Writing – original draft preparation: J.S.; Writing – review & editing: J.S., H.Y., Z.W., W.L., W.D., Z.L. and Y.D. All authors have read and agreed to the published version of the manuscript.

**Competing interests**

The authors declare that they have no conflict of interest.

**Acknowledgements**

We thank Anna M. Ukkola for her solid work in flux tower data processing, which provided a robust foundation for our research. We are also grateful for her responsiveness to our questions in using PLUMBER2. We extend our thanks to Danielle Svehla Christianson of Integrated Data Systems, Scientific Data Division, and Lawrence Berkeley National Laboratory for her prompt assistance in utilizing AmeriFlux BADM data. This work accessed several flux tower networks in

the FLUXNET community, including AmeriFlux, ChinaFlux, European Fluxes, OzFlux, and Swiss Fluxnet. Special thanks to FLUXNET and AmeriFlux for providing the BADM data. We also accessed data from ESRL's Global Monitoring Laboratory (GML) of the National Oceanic and Atmospheric Administration (NOAA) and the AT-Neu site research group. This work is based on numerous publications, we thank these scientists for sharing their data. The $PFT_{local}$ maps were supported and provided by the ESA CCI Land Cover project. The Köppen-Geiger climate classification maps are hosted by

GloH2O.

**Financial support**

This work was supported by the Natural Science Foundation of China (under Grants 42075160 and 42088101), Guangdong Major Project of Basic and Applied Basic Research (2021B0301030007), the Southern Marine Science and Engineering Guangdong Laboratory (Zhuhai) (No. SML2023SP216) and the specific research fund of The Innovation Platform for

Academicians of Hainan Province (YSPTZX202143).

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
