# Peer review of "A flux tower site attribute dataset intended for land surface modeling"

_Earth System Science Data, 2024_

## Referee Comment (RC1)

Review of manuscript MS No.: essd-2024-77

Online discussion available at: https://doi.org/10.5194/essd-2024-77

**A flux tower site attribute dataset intended for land surface modeling**

By Jiahao Shi et al.

This paper describes a dataset constructed by adding additional metadata variables to a set of existing data from flux tower observations. The additional variables added by the authors to this dataset include local vegetation properties, topographic parameters, soil properties, and wind measurement height. The purpose of this study is to provide a dataset to be used for development and validation of land surface models (LSMs). To evaluate the implication of using these updated variables with respect to default values commonly used in LSMs, the authors perform some experiments using a LSM. This analysis shows that these input variables can have significant impacts on the simulated fluxes of water and energy between land and atmosphere.

I believe the effort of enhancing existing datasets by adding additional variables and site characteristics as done here is certainly worthwhile, and therefore believe the manuscript would be of interest for the readership of the journal. The paper is overall concise and suitably organized to present the data and an application of their use in LSMs. However, the language used in the paper needs improvement. Several sentences throughout the paper should be rephrased in my opinion, as they make the text unclear and not very precise. I list several of these instances in the specific comments below, but I would recommend a thorough check of the language used in the paper. I believe the authors should address the comments below before the manuscript is considered for publication.

**Comments**

While I understand the focus of the paper is on presenting the new dataset, I believe a short description of the treatment of water and energy fluxes at the land surface in the model used here (CoLM) would be very helpful. Since the model results are an important part of the manuscript, this would help the readers in interpreting the improvement shown due to the improved data sources.

It seems that when vegetation or soil properties data are not available for the sites, the authors use the "default data" instead in order to fill the missing data. I believe the authors should provide some information about potential inhomogeneities in the final dataset resulting from this choice. If I understand correctly, the default data used to fill missing data here are those also shown in Figure 3 as comparison. I would recommend the authors use the sites for which both data sources are available to provide some quantification of the difference between new in-situ data and default data, thus quantifying the resulting inhomogeneities in the final data product. Some of this information may already been shown in Figure 3, but I recommend the authors quantify this explicitly as it is an important feature of the data produced here.

**Minor comments and (non-exhaustive) suggestions on the language used in the paper**

L20: Which model? Or do you mean "models"?

L369: "Using CoLM at 36 sites": Is there a specific reason the model was run at 36 sites out of 90 and not at all? In particular, at line 378 it is stated that all selected sites used for the modelling experiment have fairly large LAI values, but a large sensitivity to LAI is expected at sites characterized by lower LAI.

L40: maybe "for testing and validating LSMs"?

L41: it suffers -> these datasets suffer

Figure 7 caption: Do you mean "Precip" in the legend?

L117: please clarify sentence.

L142: because -> since

L142: "they are close numerically" – could you be more precise and state how similar these data sources are?

L144: what are "site pictures"? satellite imagery? Could you please specify and indicate the data source?

L209 "between RUNS using…"

L311: but -> however

L372: Remove "And"

L392: A previous study found / discovered / stated ….

L393: This study, however, …

L399: Remove "And"

Eq. (1): Is n=365 here?

---

## Referee Comment (RC4)

**Review comments**

Shi et al. present a study on a flux tower site attribute dataset intended for land surface modeling. This dataset is very valuable for land surface modeling and beyond. Overall, the manuscript is well-written, I have some additional comments for consideration.

**Specific Comments**

1. **Line 102:** "Picking years with a low gap-filled percentage for fluxes (latent and sensible heat) and vapor pressure deficit (VPD)." Why were these variables chosen for picking? Why not include precipitation, temperature, etc.? Please provide an explanation.

2. One of our studies (https://papers.ssrn.com/sol3/papers.cfm?abstract_id=4732309) conducted land surface model evaluations at the site level using the PLUMBER2 datasets, which are very valuable. We took time to find seek additional model inputs, such as soil texture information, from the literature. This study is convenient for land surface modelers. Therefore, it would be even more valuable if it included more sites and longer period of data beyond the PLUMBER2 sites.

3. Another study of ours developed global 1km land surface parameters for earth system modeling (https://essd.copernicus.org/articles/16/2007/2024/essd-16-2007-2024.html), sharing some data sources with this study, such as PFTs classification. Combining this 1km data with the site-level study could enhance land surface modeling. For example, global 1km data could provide topography attributes for sites lacking this information. This could be discussed in the manuscript.

4. **Line 92:** "Three global datasets were used to complement attribute data of sites lacking site-observed FVC, LAI, and soil texture." It's great to analyze the uncertainty by using global datasets to fill site data. How accurate are these global datasets? The authors could analyze the consistency between sites with complete attributes and the corresponding global datasets. This information would be valuable for readers to understand the uncertainties introduced by using global datasets.

5. **Line 195:** It seems not all sites have elevation, slope, and aspect information from literature. How were these attributes obtained for sites lacking them? Seems not mentioned in the manuscript.

6. **Table 2:** Should "LAI_default" be "LAI_max_default"? Similarly, for "LAI_site."

7. **Line 220:** "Resulting in six available sites. These sites were simulated to show the respective impact of different attributes in model results." This sentence is unclear. How many experiments were run for each site? This section could be written more clearly to make Section 3.3 easier to understand.

Lingcheng Li (PNNL)

---

## Author Comment (AC1)

**Reply to Referee #1' s comments**

**Title: A flux tower site attribute dataset intended for land surface modeling**
**No.: essd-2024-77**

We would like to thank you for your careful reading, helpful comments, and constructive suggestions, which have significantly improved the presentation of our manuscript.

All comments are addressed on a point-by-point basis below. The comments are laid out below in italicized font and specific concerns are numbered. Our response is given in normal font. The list of all related changes is given in blue text.

**Comment 1:** *While I understand the focus of the paper is on presenting the new dataset, I believe a short description of the treatment of water and energy fluxes at the land surface in the model used here (CoLM) would be very helpful. Since the model results are an important part of the manuscript, this would help the readers in interpreting the improvement shown due to the improved data sources.*

**Response1:** Thank you for your careful evaluation of this manuscript. Following your suggestions, we have added a description of how water and energy fluxes are treated at the land surface in the model (CoLM) used here. The comparison before and after modification is as follows:

**Origin:**

"The impact of collected attributes on carbon, water, and energy fluxes is assessed through single-point simulations using the Common Land Model (CoLM) (Dai et al., 2003). We used its latest version, CoLM202X (https://github.com/CoLM-SYSU/CoLM202X/tree/master, last access: 21 November 2023)."

**Revised:**

"The impact of collected attributes on carbon, water, and energy fluxes is assessed through single-point simulations using the latest version of the Common Land Model (Dai et al., 2003) (CoLM202X, https://github.com/CoLM-SYSU/CoLM202X/tree/master, last access: 21 November 2023). CoLM202X incorporates processes related to biogeophysics, biogeochemistry, ecological dynamics and human activities, and also offers optional processes and schemes which can be customized by the user. In our experiments, vegetation is modeled using a set of time-invariant parameters (optical properties: leaf optical properties; morphological properties: canopy height, vegetation root depth and profile, leaf size and angle distributions; and physiological properties). The dynamic vegetation module is turned off and the time-variant LAI and stem area index (SAI) values are prescribed from the reprocessed MODIS LAI data (Lin et al., 2023; Yuan et al., 2011). The two-big-leaf model (Dai et al., 2004) is employed to calculate processes such as radiative transfer (Yuan et al., 2017), photosynthesis (Collatz et al., 1992; Farquhar et al., 1980), and stomatal conductance (Ball et al., 1987). Surface turbulent exchange is simulated using similarity theory (Brutsaert, 1982; Zeng and Dickinson, 1998). Total evapotranspiration includes evaporation from stems, leaves, and the ground, as well as vegetation transpiration. Surface and subsurface runoff consider factors such as terrain, groundwater level, precipitation, and infiltration rate. Additionally, the model accounts for processes including precipitation phase and intensity, canopy

interception, vertical movement of water in snow and soil, and snow compaction (Dai et al., 2003).

***Comment 2:*** *It seems that when vegetation or soil properties data are not available for the sites, the authors use the "default data" instead in order to fill the missing data. I believe the authors should provide some information about potential inhomogeneities in the final dataset resulting from this choice. If I understand correctly, the default data used to fill missing data here are those also shown in Figure 3 as comparison. I would recommend the authors use the sites for which both data sources are available to provide some quantification of the difference between new in-situ data and default data, thus quantifying the resulting inhomogeneities in the final data product. Some of this information may already been shown in Figure 3, but I recommend the authors quantify this explicitly as it is an important feature of the data produced here.*

**Response2:** We completely agree with your suggestion. It is necessary to account for potential inhomogeneities in the final dataset resulting from data filling. Figure 3 shows the discrepancies between site data and default data to demonstrate the importance of site data. After careful consideration, we believe that Sect. 3.2 describes the flux tower site attribute dataset. Therefore, the quantification of discrepancies between site data and filled data has been added to Sect. 3.2, illustrating the inhomogeneities in the final dataset due to data filling. The added information is as follows.

One point of clarification is that the default data and the data used to fill the missing data (Filled data) are not exactly the same. The default data is the data commonly used in the LSMs. Filled data is used to fill the missing site-observed attributes in the final dataset. The details are shown in the table below:

| Attribute | Default data | Filled data | Consistency |
|---|---|---|---|
| PCT_PFT | IGBP classification | $PFT_{local}$ maps (Harper et al., 2023) | Inconsistent |
| LAI | Reprocessed MODIS 6.1 LAI (Lin et al., 2023) | | Consistent |
| Canopy height | Lookup table from CoLM | PLUMBER2 (Ukkola et al., 2022) | Inconsistent |
| Soil texture | GSDE soil dataset (Shangguan et al., 2014) | | Consistent |

**Add:**

[Figure]

**Figure 4.** Quantification of discrepancies between site data and filled data of (a) PCT_PFT, (b) maximum LAI, (c) canopy height, and (d) the percentage of sand (at all sites for which both data sources are available). The 16 PFTs were divided into three main categories (bare soil, woody, and herbage) to be quantified separately,

Figure 4 quantifies the differences between site data and filled data at all sites for which both data sources are available, illustrating the inhomogeneities in the final dataset due to data filling. The differences in vegetation cover (including bare soil, woody, and herbaceous vegetation) generally fall within 20%, with a minority of sites exceeding 40%. The mean and median LAI differences are about 1 m2/m2. Canopy height deviations are primarily within 2 m, although a few sites exceed 4 m. Differences in sand content typically remain within 30%, with both mean and median differences below 15%. The quantification indicates that the filled data are relatively reliable across most sites.

**Comment 3 (L20):** *Which model? Or do you mean "models"?*

**Response3:** Thank you for pointing this out. I'm sorry for the ambiguity. What we are trying to express here is the data commonly used by LSMs. Therefore, it should be 'models' instead of 'the model'. And we changed the wording to express it more clearly. The comparison before and after modification is as follows:

**Origin:**

"the attribute data observed at the site and the defaults of the model"

**Revised:**

"the attribute data observed at the site and commonly used by LSMs"

**Comment 4 (L369):** *"Using CoLM at 36 sites": Is there a specific reason the model was run at 36 sites out of 90 and not at all? In particular, at line 378 it is stated that all selected sites used for the modelling experiment have fairly large LAI values, but a large sensitivity to LAI is expected at sites characterized by lower LAI.*

**Response4:** Thank you for your question. In our opinion, the basis for simulation differences lies in differences in attribute values, with greater disparities in attributes values typically leading to more pronounced differences in model results. Therefore, we selected 10 sites with the largest differences between site data and default data for LAI, tree height, and soil texture, respectively. Specifically, for vegetation cover, sites with IGBP types that are a combination of tree and grasses (OSH, WSA, SAV) were chosen, resulting in six available sites. Thus, a total of 36 sites were used for modeling assessment.

In line 377, we note that variations in unit LAI elicit more substantial fluctuations in fluxes at lower LAI values (usually less than 2 m2/m2), indicating greater sensitivity of fluxes to LAI. Consider that the modeling assessment of attribute data has focused primarily on the magnitude of the impact of the attribute data and has not addressed specialized sensitivity analyses. We believe that this passage may cause some misunderstanding. Therefore, after careful consideration, we removed this part of the argument from the manuscript.

**Delete:**

Notably, unit LAI variations elicit more substantial fluctuations in fluxes at lower LAI values (usually less than 2 m2/m2), according to Launiainen et al. (2016). In light of that, all of the sites we chose have LAI values greater than 2 m2/m2, except US-GLE, the impact of LAI obtained here are relatively minor.

***Comment 5 (L40):*** *Maybe "for testing and validating LSMs"?*

**Response5:** Thanks for your suggestion. We have revised it according to your suggestion.

**Origin:**

"flux tower data was not originally designed for LSMs"

**Revised:**

"flux tower data was not originally designed for testing and validating LSMs"

***Comment 6 (L41):*** *It suffers -> these datasets suffer*

**Response 6:** Thanks for your suggestion. We have revised it according to your suggestion.

**Origin:**

"it suffers from poor data quality and a deficiency of attribute data"

**Revised:**

"these datasets suffer from poor data quality and a deficiency of attribute data."

***Comment 7 (Figure 7 caption):*** *Do you mean "Precip" in the legend?*

**Response 7:** Thank you for your careful examination. We have revised it.

**Origin:**

"Pricip."

**Revised:**

"Precip."

***Comment 8 (L117):*** *Please clarify sentence.*

**Response 8:** Thanks for the suggestion. We have described it more clearly.

**Origin:**

"The PLUMBER2 dataset got 170 sites by screening meteorological data."

**Revised:**

"The PLUMBER2 dataset got 170 sites by screening five key meteorological variables that have the largest influence on LSM simulations: incoming shortwave radiation, precipitation, air temperature, air humidity, and wind speed."

***Comment 9 (L142):*** *because -> since;*

**Response 9:** Thank you for your correction. We have revised it.

**Origin:**

"Because FVC"

**Revised:**

"Since FVC"

***Comment 10 (L142):*** *"they are close numerically" – could you be more precise and state how similar these data sources are?*

**Response 10:** Thank you for your suggestion. Here, in the absence of a description of fractional vegetation cover, the percentage of vegetation flux footprint contribution or dense forest canopy basal area is used as a proxy. These values are considered the closest numerical alternatives. Unfortunately, we don't have a reliable method or any citations to provide a precise evaluation.

However, it is clear that the fractional vegetation cover directly determines the percentage of vegetation flux footprint and dense forest canopy basal area. Theoretically, fractional vegetation cover equals the percentage of vegetation flux footprint under windless conditions; Basal area is defined as the total cross-sectional area of all stems in a stand. If the canopy width and stem cross-sectional area maintain a fixed ratio, fractional vegetation cover is equal to the percentage of dense canopy basal area.

***Comment 11 (L144):*** *what are "site pictures"? satellite imagery? Could you please specify and indicate the data source?*

**Response 11:** Thank you for your question. In this context, 'site picture' refers to photographs taken at the site and does not involve satellite images. Their sources are the flux regional network and related publications. The specific sites where pictures were used for judgment and the sources are described in Table S1. We have clarified "site picture" based on your suggestion.

**Origin:**

"we referred to site pictures to make a judgment"

**Revised:**

"we referred to site pictures (photographs taken at the site) to make a judgment"

***Comment 12 (L209, L311, L372, L392, L393, L399):*** *"between RUNS using…"; "but->however"; Remove "and"; A previous study found / discovered / stated ….; This study, however, …; Remove "And".*

**Response 12:** Thank you for your correction. We've revised these words and phrases based on your

suggestion.

**Origin:**

"between using"; "But the effects of vegetation"; "And among the four attributes"; "A previous study viewed that"; "Its study, however,"; "And the attribute data we collected"

**Revised:**

"between runs using"; "However the effects of vegetation"; "Among the four attributes"; "A previous study stated that"; "This study, however,"; "the attribute data we collected"

*Comment 13 (Eq. (1)): Is n = 365 here?*

**Response 13:** Thank you for your question. 'n' stands for the number of days in different months, with a value of 28, 30 or 31, depending on the number of days in each month. Based on your comments, we have clarified Eq. (1). The comparison before and after modification is as follows:

**Origin:**

$$MD\% = \begin{cases} \dfrac{|\frac{1}{n}\sum_{i=1}^{n}(Mod_{site,i} - Mod_{default,i})|}{\frac{1}{365}\sum_{j=1}^{365} Obs_j}, & for\ LE, H, Rn, SWup, GPP, and\ Ustar \\[3mm] \dfrac{|\frac{1}{n}\sum_{i=1}^{n}(Mod_{site,i} - Mod_{default,i})|}{\frac{1}{365}\sum_{j=1}^{365} Mod_{default,j}}, & for\ SWC\ and\ TR \end{cases} \tag{1}$$

**Revised:**

$$MD\% = \begin{cases} \dfrac{|\frac{1}{n}\sum_{i=1}^{n}(Mod_{site,i} - Mod_{default,i})|}{\frac{1}{365}\sum_{j=1}^{365} Obs_j}, & for\ LE, H, Rn, SWup, GPP, and\ Ustar \\[3mm] \dfrac{|\frac{1}{n}\sum_{i=1}^{n}(Mod_{site,i} - Mod_{default,i})|}{\frac{1}{365}\sum_{j=1}^{365} Mod_{default,j}}, & for\ SWC\ and\ TR \end{cases} \qquad n = days\ of\ month \tag{1}$$

We appreciate your warm work earnestly, and hope the correction made will meet with approval. These comments and suggestions have significantly improved the quality of our manuscript.

As you mentioned, the manuscript still contains many unclear expressions. Other reviewers have also pointed out similar issues. We are very grateful for the partial corrections you have already provided. We will thoroughly check for language problems in the manuscript before the submission after the discussion phase concludes.

**References:**

Ball, J. T., Woodrow, I. E., and Berry, J. A.: A Model Predicting Stomatal Conductance and its Contribution to the Control of Photosynthesis under Different Environmental Conditions, in: Progress in Photosynthesis Research, edited by: Biggins, J., Springer Netherlands, Dordrecht, 221–224, https://doi.org/10.1007/978-94-017-0519-6_48, 1987.

Brutsaert, W.: Energy Budget and Related Methods, in: Evaporation into the Atmosphere, Springer

Netherlands, Dordrecht, 209–230, https://doi.org/10.1007/978-94-017-1497-6_10, 1982.

Collatz, G., Ribas-Carbo, M., and Berry, J.: Coupled Photosynthesis-Stomatal Conductance Model for Leaves of C4 Plants, Functional Plant Biol., 19, 519, https://doi.org/10.1071/PP9920519, 1992.

Dai, Y., Zeng, X., Dickinson, R. E., Baker, I., Bonan, G. B., Bosilovich, M. G., Denning, A. S., Dirmeyer, P. A., Houser, P. R., Niu, G., Oleson, K. W., Schlosser, C. A., and Yang, Z.-L.: The Common Land Model, Bulletin of the American Meteorological Society, 84, 1013–1024, https://doi.org/10.1175/BAMS-84-8-1013, 2003.

Dai, Y., Dickinson, R. E., and Wang, Y.-P.: A Two-Big-Leaf Model for Canopy Temperature, Photosynthesis, and Stomatal Conductance, J. Climate, 17, 2281–2299, https://doi.org/10.1175/1520-0442(2004)017<2281:ATMFCT>2.0.CO;2, 2004.

Farquhar, G. D., Von Caemmerer, S., and Berry, J. A.: A biochemical model of photosynthetic $CO_2$ assimilation in leaves of C3 species, Planta, 149, 78–90, https://doi.org/10.1007/BF00386231, 1980.

Harper, K. L., Lamarche, C., Hartley, A., Peylin, P., Ottlé, C., Bastrikov, V., San Martín, R., Bohnenstengel, S. I., Kirches, G., Boettcher, M., Shevchuk, R., Brockmann, C., and Defourny, P.: A 29-year time series of annual 300 m resolution plant-functional-type maps for climate models, Earth Syst. Sci. Data, 15, 1465–1499, https://doi.org/10.5194/essd-15-1465-2023, 2023.

Lin, W., Yuan, H., Dong, W., Zhang, S., Liu, S., Wei, N., Lu, X., Wei, Z., Hu, Y., and Dai, Y.: Reprocessed MODIS Version 6.1 Leaf Area Index Dataset and Its Evaluation for Land Surface and Climate Modeling, Remote Sensing, 15, 1780, https://doi.org/10.3390/rs15071780, 2023.

Shangguan, W., Dai, Y., Duan, Q., Liu, B., and Yuan, H.: A global soil data set for earth system modeling, J. Adv. Model. Earth Syst., 6, 249–263, https://doi.org/10.1002/2013MS000293, 2014.

Ukkola, A. M., Abramowitz, G., and De Kauwe, M. G.: A flux tower dataset tailored for land model evaluation, Earth Syst. Sci. Data, 14, 449–461, https://doi.org/10.5194/essd-14-449-2022, 2022.

Yuan, H., Dai, Y., Xiao, Z., Ji, D., and Shangguan, W.: Reprocessing the MODIS Leaf Area Index products for land surface and climate modelling, Remote Sensing of Environment, 115, 1171–1187, https://doi.org/10.1016/j.rse.2011.01.001, 2011.

Yuan, H., Dai, Y., Dickinson, R. E., Pinty, B., Shangguan, W., Zhang, S., Wang, L., and Zhu, S.: Reexamination and further development of two-stream canopy radiative transfer models for global land modeling, J Adv Model Earth Syst, 9, 113–129, https://doi.org/10.1002/2016MS000773, 2017.

Zeng, X. and Dickinson, R. E.: Effect of Surface Sublayer on Surface Skin Temperature and Fluxes, J. Climate, 11, 537–550, https://doi.org/10.1175/1520-0442(1998)011<0537:EOSSOS>2.0.CO;2, 1998.

---

## Author Comment (AC2)

**Title: A flux tower site attribute dataset intended for land surface modeling**
**No.: essd-2024-77**

*This paper describes a dataset based on flux-tower measurements obtained from network databases, which underwent additional quality control and were combined with ancillary data characterizing the sites. The dataset was created to make flux-tower measurements including site characteristics available to the land surface modelling community, enabling site-level simulations with site-specific soil and vegetation information, where available. The additional quality control reduced the number of available sites and resulted in discontinuous time series at least at some sites. The paper shows that land surface model (LSM) simulations with soil and vegetation characteristics obtained from global gridded datasets instead of site-specific data can lead to large differences in simulated pools and fluxes.*

*I believe that a dataset including both flux-tower observations as well as site attributes required to run and evaluate LSMs is of interest to the community and useful for model development. The paper is generally well organized and written. There are, however, several sentences, which are not completely clear and should be rephrased. Generally, the paper should be checked and corrected for language issues. I have mentioned some, but not all, of these in the specific comments. I suggest that the below comments should be addressed before publication.*

Thank you for your careful evaluation of this manuscript. We greatly appreciate your positive and constructive comments on our manuscript, which have significantly improved the quality of our manuscript.

All comments are addressed on a point-by-point basis below. The comments are laid out below in italicized font and specific concerns are numbered. Our response is given in normal font. The list of all related changes is given in blue text.

**Comment 1:** *It should be made clearer what exactly the quality control entailed and whether all variables were removed from the dataset, when one of the variables was gap-filled or had lower quality data, or if just that particular variable was removed. It is not completely clear to me whether both the atmospheric forcing variables and the flux measurements used to evaluate LSMs have discontinuous timeseries in the dataset. If the forcing variables are discontinuous, the authors should make it clearer how this is handled in LSMs and how the data are still useful for LSMs.*

**Response1:** Thank you for your careful evaluation of this manuscript. We fully agree with your opinion. We have provided a more detailed description of the variables excluded by quality control. Additionally, during the screening process, we excluded all variables when one of the variables was gap-filled or had lower quality data, because the selected variables were basic, and users can still easily obtain the full variables and time series through PLUMBER2. This is a brief response, please refer to Response 16 for specific details.

Discontinuous meteorological data are indeed difficult to apply to LSMs. Therefore, we simulated all years in the PLUMBER2 dataset, but subsequent analyses are conducted exclusively for the years we have chosen. Please refer to Response 18 for specific details.

**Comment 2:** *Regarding the soil attributes that were included for the sites, I'd be interested why the authors do not mention soil depth. I'm aware that soil depths measurements are generally not available for the sites, but it is an important variable in many LSMs. Even if it is obtained from global gridded datasets, it could still be useful to include in this dataset. Another variable, which was not included, is the measurement height of air temperature. As this is required in several LSMs and is not always the same height as the measurement height of wind speed, I think it would be useful to include the air temperature measurement height as well or explain why it was not included.*

**Response 2:** We totally agree with your suggestion. We have added soil depth as well as the measurement heights of air temperature and humidity to the attribute dataset. Please see Responses 7 and 12 for the processing of soil depth, and Response 13 for the processing of measurement heights of air temperature and humidity.

**Comment 3:** *Some of the Tables and Figures could be improved by organizing sites in the same order for the different variables that are shown or to show the selected variables for all the sites. For example, Table 2 and Figure 7 could be made clearer.*

**Response 3:** Thank you for your suggestions. We apologize for the ambiguity. We have reorganized Table 2 by adding the latitude and longitude for the sites and lining up all sites in a single column. The site order is arranged according to the first letter. Please see Response 22 for changes to Table 2.

Figure 7 is intended to illustrate that the impact of attributes is substantially associated with precipitation. We intentionally chose two typical sites for each attribute and formed a contrasting effect to illustrate the important role of precipitation. Therefore, only 8 sites are ultimately shown. We have added this information to the description of Figure 7. Please refer to Response 31 for specific details.

**Comment 4 (L15):** *Be more specific what you mean with "external disturbances"? Aren't all disturbances external?*

**Response 4:** Thank you for your question. I'm sorry I didn't make it clear "external disturbances". It includes irrigation, deforestation, and water body disturbance (details in L132). The specific disturbance events for the 10 disturbed sites are shown in Table S3. We have clarified this according to your suggestion.

**Origin (L15):**

"including the proportion of gap-filled data, external disturbances, and energy balance closure (EBC),"

**Revised (L15):**

"including the proportion of gap-filled data, energy balance closure (EBC), and external disturbances such as irrigation and deforestation,"

**Comment 5 (L51):** *It should be "at some sites".*

**Response 5:** Thank you for your correction. We have revised it.

**Origin (L51):**

"in some sites"

**Revised (L51):**

"at some sites"

**Comment 6 (L55):** *For site-level simulations, it isn't always the case that gridded data products are used to obtain soil textures, etc., if site-specific information is available in the literature.*

**Response 6:** Thank you for your correction. It is true, as you say, that the attribute data used is not always globally gridded data products. We've revised the wording based on your suggestion.

**Origin (L56):**

"the current practice involves deriving these attribute data"

**Revised (L56):**

"the current practice usually obtains these attribute data"

**Comment 7 (L76):** *Why are LAI and canopy height included in the four most important attributes, even though they aren't required as inputs for many LSMs? Soil depth, however, is not mentioned, which can strongly impact model outputs and is required my many LSMs as well.*

**Response 7:** Thank you for your question. Yes, if the Dynamic Global Vegetation Model (DGVM) is not activated, LAI and canopy height are not required for LSMs. This approach is mainly suitable for long-term climate simulations. In such cases, LSMs use the canopy structural parameters from DGVM's outputs. However, in a relatively short-term simulations (i.e., weather or seasonal scale) or historical simulations (i.e., current climate 2000-2020), the DGVM is typically turned off and prescribed LAI and tree height values are used (Forzieri et al., 2020; Zeng et al., 2017). These values are generally derived from remote sensing 'observations' or in-situ measurements. This is the approach employed in our study. An important reason for this approach is that LAI and canopy height are critical vegetation structure data. In particular, LAI affects processes such as radiative transfer and surface flux exchanges. Canopy height directly determines the zero-plane displacement height and the roughness length, consequently influencing the intensity of land-atmosphere flux exchange. The LAI or canopy height simulated by DGVMs generally shows larger uncertainties compared to remote sensing or in-situ observations. That's why we collected the LAI, canopy height values in this study.

Thanks for the reminder. Soil depth is indeed an indispensable variable in LSMs. Therefore, we have added the site soil depth data to the attribute dataset. The dataset is available at https://doi.org/10.5281/zenodo.12596218. We have also updated the graphs and related text describing the attribute data.

**Origin (L187):**

**Soil bulk density and organic carbon concentration**

Soil bulk density and organic carbon concentration data are sourced from site descriptions in literature, regional networks, and AmeriFlux BADM file. Specifically, soil bulk density data were collected at 37 sites, and soil organic carbon concentration at 23 sites. At 32 and 22 sites, respectively, the observation depth was given. Despite the scarcity of site-observed data for these two soil attributes, we have included them in the final dataset. For site-specific studies, they can provide useful references for researchers.

**Revised (L197):**

**Soil bulk density, organic carbon concentration, and depth**

Soil bulk density, organic carbon concentration, and depth data are sourced from site descriptions in literature, regional networks, and AmeriFlux BADM file. Specifically, soil bulk density was collected at 37 sites, soil organic carbon concentration at 23 sites, and soil depth at 31 sites. The observation depth was provided for soil bulk density at 32 sites and for organic carbon concentration at 22 sites. Despite the scarcity of site-observed data for the three soil attributes, we have included them in the final dataset. For site-specific studies, they can provide useful references for researchers.

**Origin (Figure 2):**

[Figure]

**Revised (Figure 2):**

[Figure]

**Origin (Figur 2 caption):**

"(d) Number of collected site-observed attribute data for PCT_PFT, maximum LAI (LAI), mean canopy height ($H_{can}$), soil texture (TEX), bulk density (BD) and organic carbon concentration (OC), elevation (Elev), slope, aspect, and wind reference measurement height (H_ref)."

**Revised (Figur 2 caption):**

"(d) Number of collected site-observed attribute data for PCT_PFT, maximum LAI (LAI), mean canopy height ($H_{can}$), soil texture (TEX), bulk density (BD), organic carbon concentration (OC), and soil depth (Depth), slope, aspect, and reference measurement height (Wind: $H_v$; Air temperature: $H_t$; Humidity: $H_q$)."

**Origin (Table 3):**

| Variable (Dimension) | Long name | Unit | Description |
|---|---|---|---|
| PCT_PFT (pft=16) | Percent plant functional types cover | % | Source[a]; |
| LAI_Max | Maximum leaf area index | $m^2/m^2$ | Source; year_range[b]; LAI_Max_year[c] |
| Canopy_height | Canopy height | m | Source; |
| Soil_TEX (particle_size=3, soil_layer=4) | Soil texture(sand/silt/clay) | % | Source; layer_n_depth[d] |
| Soil_BD (soil_layer=4) | Soil bulk density | g cm$^{-3}$ | Source; layer_n_depth[d] |
| Soil_OC (soil_layer=4) | Soil organic carbon concentration | % | Source; layer_n_depth[d] |
| Elevation | Site elevation | m | Source; |
| Slope | Site slope | - | Source; |
| Aspect | Site aspect | - | Source; |
| Reference_height | Measurement height of wind speed or flux | m | Source; Measurement variable (Wind or Flux) |
| year_qc (year=21) | Selected year of high-quality data | - | - |

**Revised (Table 3):**

| Variable (Dimension) | Long name | Unit | Description |
|---|---|---|---|
| PCT_PFT (pft=16) | Percent plant functional types cover | % | Source[a]; |
| LAI_Max | Maximum leaf area index | $m^2/m^2$ | Source; year_range[b]; LAI_Max_year[c] |
| Canopy_height | Canopy height | m | Source; |
| Soil_TEX (particle_size=3, soil_layer=4) | Soil texture(sand/silt/clay) | % | Source; layer_n_depth[d] |
| Soil_depth | Soil depth | cm | Source; |
| Soil_BD (soil_layer=4) | Soil bulk density | $g\ cm^{-3}$ | Source; layer_n_depth[d] |
| Soil_OC (soil_layer=4) | Soil organic carbon concentration | % | Source; layer_n_depth[d] |
| Slope | Site slope | - | Source; |
| Aspect | Site aspect | - | Source; |
| Reference_height_v | Measurement height of wind speed or flux | m | Source; Measurement variable (Wind or Flux) |
| Reference_height_t | Measurement height of air temperature or flux | | Source; Measurement variable (Wind or Flux) |
| Reference_height_q | Measurement height of air humidity or flux | | Source; Measurement variable (Wind or Flux) |
| year_qc (year=21) | Selected year of high-quality data | - | - |

***Comment 8 (L85):*** *What are the "7 site-related articles" and why do you mention the number? It doesn't seem like you use site-specific publications for all the sites, so what is special about these 7?*

**Response 8:** Thank you for your question. These 7 site-related articles contain information on the proportions of C3/C4 grass and are therefore used for PFTs classification (including sites AU-How, PT-Mi2, SD-Dem, US-Aud, US-Fpe, US-Var, US-Wkg). We apologize for not being clearly explained for the 7 site-related articles. We have added new footnotes to Table S1 to clarify these articles and their corresponding sites.

**Origin (L85):**

"*7 site-related articles*"

**Revised (L85):**

"7 site-related articles for C3/C4 classification"

**Add (Table S1):**

"[f] Sites using literature descriptions for C3/C4 classification"

***Comment 9 (L90):*** *Better than what?*

**Response 9:** Thank you for your question. Here it means better than MODIS LAI. To provide a clearer introduction to the reprocessed MODIS LAI, we have reorganized language. The comparison before and after modification is as follows:

**Origin (91):**

"And the reprocessed MODIS LAI is much smoother and more consistent with adjacent values, displaying better spatiotemporally continuous and consistency."

**Revised (91):**

"The reprocessed MODIS LAI used the modified temporal spatial filter (mTSF) method for a simple data assimilation, then applied the post processing-TIMESAT (A software package to analyze time-series of satellite sensor data) Savitzky–Golay (SG) filter to get the final result. Site LAI validation shows that the reprocessed MODIS LAI is much smoother and more consistent with adjacent values than the original MODIS LAI, and closer to site observations (Lin et al., 2023; Yuan et al., 2011)."

*Comment 10 (L93): What exactly do you mean with "LAI complements"? Are these site measurements gap-filled with MODIS LAI?*

**Response 10:** Thank you for your question. We apologize for the ambiguity. "LAI complements" indicates site measurements gap-filled with MODIS LAI. we have changed the wording to express it more clearly.

**Origin (93):**

"LAI complements still use the reprocessed MODIS LAI. FVC complements use a global 300m PFT maps"

**Revised (93):**

"LAI filling still uses the reprocessed MODIS LAI, whereas the FVC filling employs a global 300 m PFT map"

*Comment 11 (L96): It should be "use" instead of "using". Otherwise, the sentence is incomplete.*

**Response 11:** Thank you for your correction. We have revised it.

**Origin (96):**

"Complements of soil texture using"

**Revised (96):**

"Filling of soil texture uses"

*Comment 12 (L105): Why don't the soil attributes include soil depth? That is used in many LSMs as well and can have strong impacts on soil moisture and temperatures.*

**Response 12:** Thank you for your suggestion. We agree that soil depth is indeed an important variable in LSMs. However, many LSMs currently treat soil depth in a simplistic manner, setting it to a constant value (e.g., CABLE, CoLM, Noah-MP, etc.) on a global scale. Therefore, we did not consider soil depth in the initial attribute dataset.

Considering that soil depth has strong impacts on soil moisture and temperatures, we have added the site soil depth values to the attribute dataset. The updated dataset is available at https://doi.org/10.5281/zenodo.12596218.

***Comment 13 (L108):*** *What do you mean with "revised by wind speed measurement height"? Also, why only wind speed? The measurement height of air temperature is required by many models as well and isn't always the same height as the wind speed measurement height.*

**Response 13:** Thank you for your question. There is no specific observed variable for the reference measurement height of existing flux tower dataset. Given that wind speed varies most at different heights, we use the wind speed measurement height as the reference measurement height. That is "revised by wind speed measurement height".

Indeed, as you say, the measurement height of air temperature is required by many models as well and isn't always the same height as the wind speed measurement height. We fully agree with you. So, we have added the reference measurement heights for air temperature and humidity to the attribute dataset as well. Thank you very much for your suggestion. The updated dataset is available at https://doi.org/10.5281/zenodo.12596218.

We have also updated the graphs and related text describing the attribute data in the article.

**Origin (106):**

"the reference measurement height (for emulating the lowest layer of the atmospheric model to which the LSM would be coupled) was revised by wind speed measurement height if possible."

**Revised (106):**

"We obtained the reference measurement height (for emulating the lowest layer of the atmospheric model to which the LSM would be coupled) of wind speed, air temperature, and air humidity."

**Origin (200):**

"From these sources, we look for the height of wind speed measurement or the height of instrument used to wind speed measurements (such as the wind cup)."

**Revised (200):**

"From these sources. we look for the height of wind speed/air temperature/air humidity measurement or the height of the instrument used for measurements (such as the wind cup and temperature and humidity sensor)."

**Origin (203):**

"As a result, wind observation heights are available for a total of 76 sites."

**Revised (203):**

"As a result, wind observation heights are available for a total of 76 sites, and 65 sites are available for air temperature and humidity observation heights."

***Comment 14 (L109):*** *"breakdown to" should be "broken down into"*

**Response 14:** Thank you for your correction. We have revised it.

**Origin (L109):**

"the FVC was further breakdown to different PFTs"

**Revised (L109):**

"the FVC was further broken down into different PFTs"

**Comment 15 (Table1):** *Why is the MODIS LAI dataset included in the table twice?*

**Response 15:** Thank you for your question. This is because the global LAI product needs to be used twice during attribute dataset generation. The first time is for C3/C4 classification (data usage: PFT information), which is described in L144. The second time is to fill in the LAI for missing measurements at the site. Therefore, the MODIS LAI dataset is included twice in the table.

**Comment 16 (L123):** *Did you exclude those years for both the fluxes and meteorology? Why did you not just remove the low-quality fluxes, but kept the meteorology and high-quality flux data for those time periods? To evaluate the model simulations, you do not necessarily need all flux data. Only the meteorological forcings have to be complete and they do not have to be of low quality, when some of the flux measurements are.*

**Response 16:** Thank you for your question. We excluded those years for both the fluxes and meteorology. We agree with you that more meteorological and flux observations could have been retained.

We excluded all flux data for two main reasons: (1) latent and sensible heat are the most important variables in land-atmosphere exchange and are the first variables to be assessed in land-atmosphere exchange. So, when the quality of latent and sensible heat is poor, we exclude all fluxes. (2) The period of poorer quality of observations for latent and sensible heat usually implies poorer quality of turbulent exchanges (e.g., carbon exchanges including GPP and respiration; friction velocities).

Despite these, there are still some model results that can be evaluated (e.g., the net and upward shortwave radiation). Therefore, we provide a more detailed description of each excluded year. Label whether the exclusion is due to the poor quality of flux, meteorology, or both. We will add this information to Tables S2 and S3 in the manuscript submission after the end of the Discussions. This allows the user to get more detailed data quality information and to choose simulation years and assessment variables according to individual needs. In addition, it should be noted that the attribute dataset only provides the results of the quality screening, and the user can still easily obtain the full variables and time series through PLUMBER2.

As you mentioned, the flux data used for evaluation does not need to be continuous. We fully agree that this approach maximizes the utilization of available data. Here, we adopted a stricter criterion by filtering the flux data annually, which enhances user convenience. Many studies also apply annual criteria for data selection. Finally, if users require flux observations for specific periods, they can easily obtain the full time series with corresponding QC flags from PLUMBER2.

***Comment 17 (L132):*** *What do you mean with "impacted by a sizable body of water"? Was the site flooded or did a lake or so develop at the site?*

**Response 17:** Thank you for your question. Here, "impacted by a sizable body of water" means "This site is unusual: it is situated on a low-lying narrow spit of land between a small lake and the Mediterranean Sea and is likely heavily influenced by horizontal advection" (Haughton et al., 2016). I'm sorry for the ambiguity. We have added reference sources here.

**Origin (L132):**

"such as irrigation, deforestation, and one site impacted by a sizable body of water"

**Revised (L132):**

"such as irrigation, deforestation, and one site impacted by a large body of water nearby (details in Table S3)"

***Comment 18 (L132):*** *"we preserved non-consecutive years that met the criteria" - Does this apply to both the meteorology as well as fluxes? As the meteorology is needed to force LSMs, using discontinuous years of meteorological data seems like it would not be very useful for LSMs and could cause crashes or strange behaviour in models, if the meteorology suddenly shifts with jumps in time. The end of one year could be much colder/warmer or wetter/drier than the beginning of the next available year, which would likely cause the model state to be out of phase with the actual meteorological conditions. Why did you decide on this approach? Also, why not include high-quality gap-filled data at least for the meteorological forcings. For the fluxes, which are only used to evaluate the models, it seems reasonable to only keep measured values, but that does not mean that the meteorology has to be discarded as well.*

**Response 18:** Thank you for your question. "we preserved non-consecutive years that met the criteria"——this applies to both the meteorology as well as fluxes.

Discontinuous meteorological data are indeed difficult to apply to LSMs. Therefore, we simulated all years in the PLUMBER2 dataset (details in L222). The meteorological data for these years are relatively reliable, except that the specific humidity at some of the sites was not thoroughly quality screened (details in L118).

High quality gap-filled data are necessary. Therefore, our quality screening considered data with high quality gap-filled data. For fluxes, data with QC = 1 were considered (details in L123). For meteorological variables, we followed PLUMBER2 and kept a smaller proportion of gap-filled data (details in L126). This is because specific humidity is one of the five variables (including incoming shortwave radiation, precipitation, air temperature, air humidity, and wind speed) that have strong impacts on LSMs (Ukkola et al., 2022).

The attribute dataset provides the results of the quality screening. As described in Response 16, we provide more detail on the excluded years, and the user still has access to all of the variables with the time series (including the meteorological variables) according to their needs.

***Comment 19 (L224):*** *Why only the first year and not all available years at the sites? One year could be*

*an unusual/extreme year and not representative of the usual site conditions.*

**Response 19:** Thank you for your suggestion. It's true that using only one year of data for SPIN-UP is not quite right. Therefore, we have changed the SPIN-UP approach. The new scheme loops the atmospheric forcing data for each site's observation period until it reaches 50 years.

We redrew the pictures associated with the model results (including Figure 4, Figure 5, Figure 6, and Figure 7). Overall, these figures don't change much. And the relevant MD% values in this paper are modified. The comparison before and after modification is as follows:

**Origin (Figure 4):**

[Figure]

**Revised (Figure 4):**

[Figure]

**Origin (Figure 5):**

[Figure]

**Revised (Figure 5):**

[Figure]

**Origin (Figure 6):**

[Figure]

**Revised (Figure 6):**

[Figure]

**Origin (Figure 7):**

[Figure]

**Revised (Figure 7):**

[Figure]

***Comment 20 (L225):*** *Why only do such a short spin-up, if GPP is evaluated as well? Are the vegetation and soil C pools prescribed and not dynamic?*

**Response 20:** Thank you for your question. We apologize for the ambiguity. In our experiments, There is no C pools simulation here, the dynamic vegetation module is turned off and the time-variant LAI and stem area index (SAI) values are prescribed from the reprocessed MODIS LAI data. Therefore, we performed a relatively short spin-up.

Based on your comments, we have added a description of the model.

**Add (L208):**

"CoLM202X incorporates processes related to biogeophysics, biogeochemistry, ecological dynamics and human activities, and also offers optional processes and schemes which can be customized by the user. In our experiments, vegetation is modeled using a set of time-invariant parameters (optical properties: leaf optical properties; morphological properties: canopy height, vegetation root depth and profile, leaf size and angle distributions; and physiological properties). The dynamic vegetation module is turned off and the time-variant LAI and stem area index (SAI) values are prescribed from the reprocessed MODIS LAI data (Lin et al., 2023; Yuan et al., 2011). The two-big-leaf model (Dai et al., 2004) is employed to calculate processes such as radiative transfer (Yuan et al., 2017), photosynthesis (Collatz et al., 1992; Farquhar et al., 1980), and stomatal conductance (Ball et al., 1987). Surface turbulent exchange is simulated using similarity theory (Brutsaert, 1982; Zeng and Dickinson, 1998). Total evapotranspiration includes evaporation from stems, leaves, and the ground, as well as vegetation transpiration. Surface and subsurface runoff consider factors such as terrain, groundwater level, precipitation, and infiltration rate. Additionally, the model accounts for processes including precipitation phase and intensity, canopy interception, vertical movement of water in snow and soil, and snow compaction (Dai et al., 2003)."

***Comment 21 (L228):*** *What do you mean here? It seems like the sentence is incomplete. Is the next sentence supposed to be part of this sentence?*

**Response 21:** Thank you for your correction. The next sentence is part of this sentence. We have revised it according to your suggestion.

**Origin (229):**

"on climate-related variables (Dirmeyer, 2011; Forzieri et al., 2020). We designed a statistical indicator called the percentage of mean difference (MD %) (Eq. 1), which is calculated as"

**Revised (229):**

"on climate-related variables (Dirmeyer, 2011; Forzieri et al., 2020), we designed a statistical indicator called the percentage of mean difference (MD %) (Eq. 1). The indicator is calculated as"

***Comment 22 (Table 2):*** *Why do you show the different attributes for different sites? Wouldn't it make more sense to select the same sites and same order of sites in the table for all attributes? Then, you also only need the site column once and it's less confusing. Regarding soil texture: are the values averages*

*over different depth or values for the top layer/near-surface?*

**Response 22:** Thanks for your question. "Show the different attributes for different sites" is because different sites were selected to show simulation differences for different attributes (see L216 for details on site selection methods). However, we have reorganized Table 2 by adding the latitude and longitude for the sites and lining up all sites in a single column. The site order is arranged according to the first letter.

For soil texture, these values are for the top layer/near-surface. We have added this information to the Table 2 (footnote 'c').

**Origin (Table 2):**

| Site | LAI_default[a] (m²/m²) | LAI_site[b] (m²/m²) | Site | TEX_default[c] | TEX_site[b] |
|------|------|------|------|------|------|
| US-KS2 | 6.6 (2005[e]) | 2.7 (2005) | IT-Cpz | 33/45/22 | 87/8/5 |
| DK-Lva | 3.1 (2004) | 6.9 | DE-Gri | 52/29/20 | 10/81/9 |
| DE-Bay | 3.6 | 6.5 | FI-Sod | 52/25/20 | 92/5/3 |
| US-Goo | 4.5 | 2.0 | ES-LMa | 49/24/24 | 80/11/9 |
| DE-Seh | 3.2 (2009) | 5.9 (2009) | AU-Cpr | 64/18/18 | 94/4/2 |
| US-GLE | 1.5 | 3.8 | SD-Dem | 67/18/14 | 96/4/0 |
| US-Moz | 6.1 (2006) | 4.0 (2006) | CZ-wet | 39/37/32 | 10/85/5 |
| DE-Gri | 6.5 (2004) | 4.4 (2004) | AU-DaP | 63/18/19 | 92/5/3 |
| IT-Cpz | 5.4 | 3.5 | AU-DaS | 63/12/25 | 92/5/3 |
| US-MMS | 7.0 | 5.2 | IT-SRo | 69/17/15 | 95/4/1 |
| **Site** | $H_{can}$\_default[d] (m) | $H_{can}$\_site[b] (m) | **Site** | **IGBP** | **PCT_PFT_site[b]** |
| IT-Cpz | 35 | 14.3 | AU-How | WSA | EBT_Tr/DBS_Te/C4 : 50/25/25 |
| BE-Vie | 17 | 33.7 | ES-LMa | SAV | EBT_Te/C3 : 20/80 |
| AU-Lit | 35 | 20.0 | SD-Dem | SAV | EBT_Tr/C3/C4 : 10/27/63 |
| DE-Hai | 20 | 33.9 | US-SRM | WSA | DBS_Te/C3/C4 : 35/43/22 |
| IT-Ren | 17 | 29.0 | US-Ton | WSA | EBT_Te/C3 : 40/60 |
| DE-Tha | 17 | 28.4 | US-Whs | OSH | Bare/DBS_Te/C3 : 39/51/10 |
| IT-Lav | 17 | 28.0 | | | |
| US-Ton | 20 | 9.9 | | | |
| RU-Fyo | 17 | 26.3 | | | |
| CH-Dav | 17 | 25 | | | |

[a] The maximum LAI at the pixel containing the site provided by Reprocessed MODIS version 6.1 LAI.

[b] Site-observed data collected in this study. [c] Soil texture (sand/silt/clay) at the site location extracted from the GSDE dataset. [d] Canopy height of the dominant vegetation type at the site from the CoLM lookup table. [e] Specific year of maximum LAI.

**Revised (Table 2):**

| Site_LAI | Lat | Lon | LAI_default[a](m²/m²) | LAI_site[b](m2/m2) |
|------|------|------|------|------|
| DE-Bay | 54.142 | 11.867 | 3.6 | 6.5 |
| DE-Gri | 50.949 | 13.512 | 6.5 (2004) | 4.4 (2004) |
| DK-Lva | 55.683 | 12.083 | 3.1 (2004) | 6.9 (2004) |
| DE-Seh | 58.871 | 6.449 | 3.2 (2009) | 5.9 (2009) |
| IT-Cpz | 41.706 | 12.376 | 5.4 | 3.5 |
| US-GLE | 41.366 | -106.24 | 1.5 | 3.8 |

| | Lat | Lon | | |
|---|---|---|---|---|
| US-Goo | 34.254 | -89.873 | 4.5 | 2.0 |
| US-KS2 | 28.605 | -80.671 | 6.6 (2005[e]) | 2.7 (2005) |
| US-MMS | 39.323 | -86.413 | 7.0 | 5.2 |
| US-MOz | 38.744 | -92.200 | 6.1 (2006) | 4.0 (2006) |

| Site_TEX | Lat | Lon | TEX-default[c] | TEX_site[b] |
|---|---|---|---|---|
| AU-Cpr | -34.002 | 140.589 | 64/18/18 | 94/4/2 |
| AU-DaP | -14.063 | 131.318 | 63/18/19 | 92/5/3 |
| AU-DaS | -14.159 | 131.388 | 63/12/25 | 92/5/3 |
| CZ-wet | 49.024 | 14.770 | 39/37/32 | 10/85/5 |
| DE-Gri | 50.949 | 13.512 | 52/29/20 | 10/81/9 (0-23cm) |
| ES-LMa | 39.942 | -5.773 | 49/24/24 | 80/11/9 (0-30cm) |
| FI-Sod | 67.361 | 26.637 | 52/25/20 | 92/5/3 |
| IT-Cpz | 41.706 | 12.376 | 33/45/22 | 87/8/5 (0-10cm) |
| IT-SRo | 43.727 | 10.284 | 69/17/15 | 95/4/1 (10-20cm) |
| SD-Dem | 13.282 | 30.478 | 67/18/14 | 96/4/0 |

| Site_HTOP | Lat | Lon | $H_{can\_}$default[d] (m) | $H_{can\_}$site[b] (m) |
|---|---|---|---|---|
| AU-Lit | -13.179 | 130.794 | 35 | 20.0 |
| BE-Vie | 50.305 | 5.998 | 17 | 33.7 |
| CH-Dav | 46.815 | 9.855 | 17 | 25 |
| DE-Hai | 51.079 | 10.453 | 20 | 33.9 |
| DE-Tha | 50.936 | 13.566 | 17 | 28.4 |
| IT-Cpz | 41.706 | 12.376 | 35 | 14.3 |
| IT-Lav | 45.956 | 11.281 | 17 | 28.0 |
| IT-Ren | 46.586 | 11.433 | 17 | 29.0 |
| RU-Fyo | 56.461 | 32.922 | 17 | 26.3 |
| US-Ton | 38.431 | -120.966 | 20 | 9.9 |

| Site_FVC | Lat | Lon | IGBP | PCT_PFT_site[b] |
|---|---|---|---|---|
| AU-How | -12.495 | 131.149 | WSA | EBT_Tr/DBS_Te/C4 : 50/25/25 |
| ES-LMa | 39.942 | -5.773 | SAV | EBT_Te/C3 : 20/80 |
| SD-Dem | 13.282 | 30.478 | SAV | EBT_Tr/C3/C4 : 10/27/63 |
| US-SRM | 31.821 | -110.866 | WSA | DBS_Te/C3/C4 : 35/43/22 |
| US-Ton | 38.431 | -120.966 | WSA | EBT_Te/C3 : 40/60 |
| US-Whs | 31.743 | -110.052 | OSH | Bare/DBS_Te/C3 : 39/51/10 |

[a] The maximum LAI at the pixel containing the site provided by Reprocessed MODIS version 6.1 LAI.

[b] Site-observed data collected in this study. [c] The top layer soil texture (sand/silt/clay) at the site location extracted from the GSDE dataset. [d] Canopy height of the dominant vegetation type at the site from the CoLM lookup table. [e] Specific year of maximum LAI.

**Comment 23 (Figure 2):** *Don't you mean "number of years", not "site numbers" in the caption for (b)? In (d), is this the actual number of sites or the percentage? The name Hcan is a little confusing, as you talk about sensible heat flux as H above and here H is height.*

**Response 23:** Thanks for your suggestion. In Figure 2 (b), the vertical coordinate is the number of sites, and the horizontal coordinate is the number/length of years.

H (sensible heat) and canopy height ($H_{can}$) do tend to be confusing, so we plan to change the abbreviations for sensible heat and latent heat to Qh and Qle in the manuscript submission after discussions, and the abbreviation for canopy height will remain the same. Thank you very much for your suggestion.

*Comment 24 (L269): Didn't you say that you excluded sites with only one year of data? How can the individual site observations range from 1 to 17 years then?*

**Response 24:** Thanks for your question. We performed a three-step screening process. First, we excluded sites with only one year of observations, as these observations may be unstable. After that, we performed fluxes and VPD screening (details in L122), which may result in some sites meeting the criteria with only one year of observations. Therefore, the range of observations for individual sites varied from 1 to 17 years.

*Comment 25 (Figure 3): I do not see the difference between site and default data for the PCT_PFT. Where is it? This also applies to l. 282. If you have multiple PFTs at the site, is the canopy height the maximum height, the average or an average weighted by the fractions of those PFTs present at the site? The same question also applies to the LAI.*

**Response 25:** Thanks for your question. For PCT_PFT, the default data uses the IGBP classifications (i.e., a single ecosystem type, such as evergreen broadleaf forest (EBF)); the site data is composed of different plant functional types (PFTs). In Figure 3 (a), the asterisk indicates that the site vegetation cover has multiple PFTs, offering a more accurate representation of the vegetation conditions compared to the IGBP classifications.

Due to the availability of data sources, site canopy heights and LAI have not reached the level of PFTs. For the default LAI, the grid LAI is given here, not the LAI of the PFTs. For the default canopy height, we provide the height of the dominant PFT (highest percentage coverage).

In addition, based on your comments, we decided to add an explanation in Section 2.3 about using site data, detailing how these site attributes were applied in the simulations. The added information is as follows:

**Add (L225):**

"In the site data simulations, we scaled the default LAI time series using maximum LAI, corrected the default canopy height using site canopy height, and replaced the default topsoil texture (0-28.9 cm) with site soil texture. For sites with multiple PFTs, we calculated the LAI for each PFT using growing degree days and PCT_PFT (Lawrence and Chase, 2007). Canopy height was classified into three groups based on PFT (trees, shrubs, or grassland), with site data used to adjust the default values for the corresponding group, while the other two groups retained their default values."

*Comment 26 (L285):* This should be "at certain sites".

**Response 26:** Thank you for your correction. We have revised it.

**Origin (L285):**

"in certain sites"

**Revised (L285):**

"at certain sites"

*Comment 27 (L292):* Rephrase this sentence to make it clearer. Do you mean the file "provides" and what do you mean with "range of years for maximum LAI"?

**Response 27:** Thank you for your correction and suggestion. We have rephrased this sentence to make it clearer. The comparison before and after modification is as follows:

**Origin (L292):**

"For the maximum LAI, the file furnishes the range of years for maximum LAI, and the maximum for a specific year."

**Revised (L292):**

"For the maximum LAI, the file provides the year range covered by maximum LAI, and the maximum for a specific year."

*Comment 28 (Table 3):* Regarding the Reference height: What about the measurement height of air temperature? That is required by some models as well. It's unclear what you mean with "b Range of years with maximum LAI". If there are multiple LAI measurements, isn't each measurement for a specific year? Otherwise, if it is the maximum LAI of a timeseries, you should make that clearer.

**Response 28:** Thank you for your suggestion. As mentioned in comment 13, we have added the reference measurement heights for air temperature and humidity to the attribute dataset.

At some sites, the maximum LAI was reported in different years. Therefore, we used "range of years of maximum LAI" to express it. As per your suggestion, we have modified it to "the year range covered by maximum LAI."

**Origin (Table 3):**

"[b] Range of years with maximum LAI."

**Revised (Table 3):**

"[b] The year range covered by maximum LAI."

*Comment 29 (L318):* It is unclear to me what you mean with "were comparatively equilibrated".

*Rephrase this to make it clearer.*

**Response 29:** Thank you for your suggestion. I am sorry for the unclear expression. We have reorganized the language.

**Origin (L317):**

"On average, the impacts of four attributes—PCT_PFT, LAI, canopy height, and soil texture—on LE and H were comparatively equilibrated."

**Revised (L317):**

"On average, changes in latent and sensible heat are not dominated by certain attributes. All four attributes —PCT_PFT, LAI, canopy height, and soil texture—have a relatively strong impact on both."

**Comment 30 (L319):** *"relatively significant" -> Do you mean it is "statistically significant"?*

**Response 30:** Thank you for your question. We realize that "relatively significant" may not be appropriate. What we are trying to express here is "relatively greater".

**Origin (L319):**

"And the effect of soil texture on LE is relatively significant"

**Revised (L319):**

"And the effect of soil texture on LE is relatively greater"

**Comment 31 (Figure 7):** *Why do you show SWup and GPP at 2 sites only and don't show the LE and H there? Also, it doesn't seem to show observations for SWup at US-KS2. Why show that variable at that site, if observations were not available? Why were these specific 8 sites chosen for the figure (and not all 36 sites) and why don't you show LE, H, GPP and SWup at all the selected sites?*

**Response 31:** Thank you for your question. I'm sorry for the confusion. In the modeling assessment of attribute data, four attributes were selected, and Figure 7 shows two typical sites for each attribute (which can be contrasted to highlight the important role of precipitation). We have added this information to the description of Figure 7.

The US-KS2 and US-GLE sites are used to illustrate the role of precipitation in the impact of LAI on model results. Specifically, the result of SWup is more convincing, so we have co-displayed SWup at the US-KS2 and US-GLE sites. Although the US-KS2 site does not have SWup observations, we can still see the difference in the simulations between the site data and the default data.

Figure 7 is intended to illustrate that the impact of attributes is substantially associated with precipitation. We intentionally chose two typical sites for each attribute and formed a contrasting effect to illustrate the important role of precipitation. Therefore, only 8 sites are ultimately shown.

**Origin (Figure 7 caption):**

"and GPP at 8 selected sites"

**Revised (Figure 7 caption):**

"and GPP at 8 selected sites (two sites for each attribute for comparison. PCT_PFT: AU-How and SD-Dem; LAI: US-KS2 and US-GLE; $H_{can}$: IT-Cpz and BE-Vie; Soil texture: FI-Sod and AU-Cpr)"

**Comment 32 (L355):** *I think it would be good to be more specific what exactly you mean here, as for example different land surface modelling groups pay attention to the site-specific data required to set up sites and many measurement groups collect at least some of the data, but it's not always easily accessible. I think it would be important to point out the need for more site attribute data to be included in flux datasets, etc.*

**Response 32:** We agree with you, and more importantly point out the need to include more site attribute data in the flux dataset. We have reformulated this sentence. The comparison before and after modification is as follows:

**Origin (L355):**

"In land surface community, flux tower attribute data is currently not given enough attention."

**Revised (L355):**

"In land surface community, flux tower attribute data is currently not given enough attention. However, the site attribute data is almost as important as the flux tower observations themselves. We hope that future flux tower datasets will provide standardized site attributes."

**Comment 33 (L369):** *Why was the model run at only 36 of the sites and how were these sites selected?*

**Response 33:** Thank you for your question. We selected sites with certain differences between site data and default data for each attribute, and finally got 36 sites. The specific method is as follows (L216): We chose ten sites for each of the attributes—LAI, canopy height, and soil texture—where site data differ the most from default data (In the lookup table canopy height simulations, sites with zero plane displacement exceeding reference measurement height are excluded.). For PCT_PFT analyses, sites with IGBP types that are a combination of trees and grasses (OSH, WSA, SAV) were chosen, resulting in six available sites. These sites were simulated to show the respective impact of different attributes in model results. As a result, 36 sites ended up being used for simulations.

**Comment 34 (L375):** *Couldn't this also be related to other uncertainties such as in soil textures, soil moisture, thermal and hydraulic conductivities, LAI and GPP affecting canopy evaporation and transpiration? Why focus on the IGBP classification?*

**Response 34:** Thank you for your question and suggestion. We did lack consideration and only focused on the IGBP classifications (which is also part of the model uncertainties). This result is indeed related to the uncertainties of the model itself as well as other input data. Based on your suggestion, we have modified this sentence. The comparison before and after modification is as follows:

**Origin (L375):**

"This may be related to the model's previous development and evaluation, which was mostly centered on the IGBP classifications"

**Revised (L375):**

"This may be related to the uncertainties of the model itself as well as other input data. Such as the vegetation biophysical parameters, soil thermal and hydraulic conductivities, etc."

*Comment 35 (L376): What do you mean with "unit LAI variations"?*

**Response *35*:** Thank you for your question and suggestion. "unit LAI variations" means a change in LAI value of 1 m2/m2.

However, according to the comments of reviewer 1, we think the modeling assessment of attribute data has focused primarily on the magnitude of the impact of the attribute data, rather than sensitivity analyses. We believe that this passage may cause some misunderstanding. Therefore, after careful consideration, we removed this part of the argument from the manuscript.

**Delete (L376):**

Notably, unit LAI variations elicit more substantial fluctuations in fluxes at lower LAI values (usually less than 2 m2/m2), according to Launiainen et al. (2016). In light of that, all of the sites we chose have LAI values greater than 2 m2/m2, except US-GLE, the impact of LAI obtained here are relatively minor.

*Comment 36 (L378): Why did you choose sites with LAI > 2 m2/m2, if the impact is larger at sites with lower LAI? As I'm not sure what you mean with "unit LAI variations", I might be misunderstanding this though.*

**Response *36*:** Thank you for your question. Although we have removed the relevant expression (Response *35*), we feel it is still necessary to explain it clearly to you.

In line 377, we noted that variations in unit LAI elicit more substantial fluctuations in fluxes at lower LAI values (usually less than 2 m2/m2). indicating greater sensitivity of fluxes to LAI. However, this does not imply that their simulation differences are greater. Therefore, we prioritized sites with larger differences in LAI values for modeling assessment (L216).

*Comment 37 (L393): Which site attributes did they modify and to what extent? What kind of site were they looking at? Also, this might be model specific how sensitive the model is to certain variables. Instead of "a previous study viewed", do you mean "showed"?*

**Response *37*:** Thank you for your question. This study modified the site's soil texture, LAI, and canopy height. Specific numerical changes can be viewed from Table 2 of Ménard et al. (2015). Measurements against which the ensemble was evaluated were collected in two sites situated 60m from one another and describing two land-cover types: one artificial forest clearing and one forest site. Variables assessed in the study included latent heat, sensible heat, soil temperature and moisture, and snow water equivalent. The authors concluded that "differences in ancillary data (attribute data) have little effect on model

results".

Based on our experience in the modeling assessment of attribute data, we believe that the model's sensitivity to different variables changes the magnitude of the quantified values, but not enough to change the main conclusions.

Thank you for your correction. We have revised "A previous study viewed". The comparison before and after modification is as follows:

**Origin (L393):**

"A previous study viewed that"

**Revised (L393):**

"A previous study stated that"

*Comment 38 (L397): "Mostly during the growing season" -> This depends. For example albedo differences due to PFT selection can have significant impacts when snow is present (depending on whether snow covers the vegetation or not, etc.).*

**Response 38:** Thank you for your suggestion. We've revised these words and phrases based on your suggestion.

**Origin (L396):**

"the impacts of attribute data on climate-related variables occur mostly during the growing season."

**Revised (L396):**

"the impact of attribute data on climate-related variables is generally over a period (mostly during the growing season) rather than throughout the year"

*Comment 39 (L402): How exactly are these low-cost? That seems to depend on whether the measurements are already done at a site or not. Especially, measurements that have to be done manually instead of automated can be labor-intensive and thus not inexpensive.*

**Response 39:** Thank you for your question. We realized we weren't making it clear. It does depend on whether site measurements have been completed. However, the attribute data used in this paper is time-invariant. Only one measurement is required, so these measurements are low-cost.

We have refined this expression to avoid confusion. The comparison before and after modification is as follows:

**Origin (L402):**

"These collections of site attributes are low-cost but would strongly benefit model enhancement."

**Revised (L402):**

"These observations and collections of site time-invariant attributes are generally low-cost but

would strongly benefit model enhancement."

**Comment 40 (L405):** *Why do you make the statement that an increasing array of surface parameters elevates the model to a heightened level of sophistication? New processes and more complexity do not necessarily improve results and increase uncertainty, as many parameter values are not well defined or constrained.*

**Response 40:** Thank you for your question. We fully agree with you. New processes and more complexity do not necessarily improve results and increase uncertainty. Therefore, these parameters must be clarified.

We apologize for the misunderstanding caused by our wording. We have revised it.

**Origin (L405):**

"sophistication"

**Revised (L405):**

"complexity"

We would like to thank you for your professional review work, constructive comments, and valuable suggestions on our manuscript. We hope the correction made will meet with approval. These comments and suggestions have significantly improved the quality of our manuscript.

As you indicated, the manuscript still has several unclear expressions. Similar issues have been noted by other reviewers as well. We greatly appreciate the partial corrections you have already provided. After the discussion phase concludes, we will thoroughly review the manuscript for language issues before submission. Once again, thank you very much for the comments and suggestions.

**References:**

[revised manuscript text omitted]

---

## Author Comment (AC3)

**Reply to Referee #3's comments**

**Title: A flux tower site attribute dataset intended for land surface modeling**
**No.: essd-2024-77**

*Shi et al. improve an existing flux tower dataset developed for land modelling. These efforts are very valuable for the land modelling community and as such it was a pleasure to read this paper. I fully agree with the authors on the need to provide improved ancillary data for modelling and commend the authors' efforts in collating data on key variables which is not a simple task. This is a valuable contribution to the field, but I do have a few comments I would ask the authors to consider:*

Thank you for your careful evaluation of this manuscript. We greatly appreciate your positive and constructive comments on our manuscript, which have significantly improved its quality.

All comments are addressed on a point-by-point basis below. The comments are presented in italicized font and specific concerns are numbered. Our response is given in normal font. The list of all related changes is given in blue text.

**Comment 1:** *I feel this paper is somewhat a missed opportunity by applying the quality control process to PLUMBER2 rather than taking the PLUMBER2 framework and applying it to newer releases of flux tower data. The datasets used in PLUMBER2 are now quite out-of-date and it would have been fantastic to see an update that incorporates newer data and possibly additional sites (e.g. from ICOS and data from individual networks)*

**Response 1:** Thank you for your careful evaluation of this manuscript. We acknowledge the importance of including additional sites such as those from ICOS and individual networks. This has also given us new ideas.

PLUMBER2 includes the major datasets since the release of flux tower data. However, since flux tower datasets like FLUXNET2015 have not been released in new versions, PLUMBER2 does not contain observational data updated in recent years. Nevertheless, the site data included in PLUMBER2 continues to serve as a valuable resource for future research.

Our work also provided a framework and makes it convenient to add more sites. All the code and related data are publicly available. New flux tower data and sites will continue to be updated in the future. This is indeed a valuable direction for future research.

**Comment 2:** *I would be cautious to only provide one LAI product. In the PLUMBER2 paper we found very large differences in LAI from the MODIS and Copernicus products at some sites. A comparison to max LAI is provided in the paper but the time evolution of MODIS and Copernicus can also be very different. I would strongly encourage the authors to also consider alternative LAI products. It was also unclear how the MODIS data was processed?*

**Response 2:** Thank you very much for your suggestions. We fully acknowledge that there can be significant differences among various satellite LAI products, as demonstrated by numerous LAI products and papers.

We have chosen only the reprocessed MODIS LAI product in our study. An important reason is that the reprocessed MODIS LAI product has been extensively validated and applied in many studies (especially within the land modeling community), making it more familiar to researchers. Thus, it serves as a valuable reference. Additionally, to show the uncertainty of the reprocessed MODIS LAI, we have quantified it using site LAI (Response 13), which will offer readers a reference for understanding the uncertainty involved.

We apologize for the lack of description regarding the reprocessed MODIS LAI. Revisions have been made based on your suggestions.

**Origin (L89):**

"And the reprocessed MODIS LAI is much smoother and more consistent with adjacent values, displaying better spatiotemporally continuous and consistency."

**Revised (L89):**

"The reprocessed MODIS LAI used the modified temporal spatial filter (mTSF) method for a simple data assimilation, then applied the post processing-TIMESAT (A software package to analyze time-series of satellite sensor data) Savitzky–Golay (SG) filter to get the final result. Site LAI validation shows that the reprocessed MODIS LAI is much smoother and more consistent with adjacent values than the original MODIS LAI, and closer to site observations (Lin et al., 2023; Yuan et al., 2011)."

**Comment 3 (L13):** *Can you mention an example here of what you mean by "site-observed attribute data"?*

**Response 3:** Thank you for your valuable feedback. We have added examples to illustrate "site-observed attribute data".

**Origin (L13):**

"More importantly, these datasets lack site-observed attribute data, limiting their use as benchmarking data."

**Revised (L13):**

"More importantly, these datasets lack site-observed attribute data, such as fractional vegetation cover and leaf area index, which limits their utility as benchmarking data."

**Comment 4 (L18):** *Please check grammar here, wording unclear*

**Response 4:** Thank you for your correction. We have corrected the grammar.

**Origin (L17):**

"Then we obtained the final flux tower attribute dataset by global data product complement and

plant functional types (PFTs) classification."

**Revised (L17):**

"We then obtained the final flux tower attribute dataset by filling in missing data with global data products and classifying plant functional types (PFTs)"

*Comment 5 (L41): would be good if you could mention some examples of "poor quality data" and "deficiency of attribute data" to make this a bit more concrete*

**Response 5:** Thank you for your suggestions. Here, we use "poor quality data" and "deficiency of attribute data" to lead into the following two paragraphs. The following two paragraphs are specific introductions of "poor quality data"and "deficiency of attribute data", respectively. We are unsure if this approach is logically coherent. If you have any other ideas, please feel free to share your suggestions.

*Comment 6 (L53): The reason for not screening flux data for gapfilling was that the requirements around this can be very study-specific. Some research questions might need high quality multi-year records whereas others might concentrate on individual events. This is just a comment but screening for flux gapfilling is challenging when creating a dataset for general use.*

**Response 6:** We fully agree with your opinion. Other reviewers have also suggested retaining more data and years. In response to this, we provide a more detailed description of each excluded year, labeling whether the exclusion is due to the poor quality of flux, meteorological variable, or both.

We will add this information to Tables S2 and S3 in the manuscript submission after the Discussions. This will allow users to get more detailed data quality information and to choose simulation years and assessment variables according to their individual needs. Users can easily obtain the complete set of variables and time series from PLUMBER2.

*Comment 7 (L55): Yes this is often the case but there are also many studies relying on site-specific information where this is available. Often this is done out of necessity as you point our on L67*

**Response 7:** Thank you for your correction. It is true, as you say, that there are also many studies relying on site-specific information where this is available. We've revised the wording based on your suggestion.

**Origin (L56):**

"the current practice involves deriving these attribute data"

**Revised (L56):**

"the current practice usually involves deriving these attribute data"

*Comment 8 (L75): No argument that these are important but can we really state that they are the most important attributes?*

**Response 8:** Thank you for your suggestion. We agree that determining the "most important" attributes can be subjective and may vary depending on the specific context and criteria used for evaluation. To provide a more balanced view, we have revised the text to express that these data are fundamental attributes needed for modeling.

**Origin (L75):**

   "Furthermore, by modeling for the four most important attribute variables—percentage of plant functional type (PFT) cover (PCT_PFT), LAI, canopy height and soil texture—we demonstrate how the site-observed attribute data and the conventional attribute data used by LSMs differ in their model output. These results emphasize the non-negligible impact of flux tower attribute data in model simulation and its development."

**Revised (L75):**

   "Furthermore, through modeling comparison for the four attribute variables—percentage of plant functional type (PFT) cover (PCT_PFT), LAI, canopy height, and soil texture—we demonstrate how the site-observed attribute data and the default attribute data used by a LSM differ in their outputs. These results emphasize the impact of flux tower attribute data on model simulations and development."

***Comment 9 (L87):*** *It is not clear how the Köppen-Geiger classification helps with LSM modelling?*

**Response 9:** Thank you for your suggestion. In this study, the 16 PFT classifications include climate types of vegetation (e.g., tropical, temperate, and boreal). The Köppen climate classification corresponds to these vegetation climate types and is therefore used for PFT classifications. Here (L87), we mainly introduce the data used without the specific methods. However, we have provided a more detailed description of the methods (Response 25).

***Comment 10 (L90):*** *MODIS v6.1 might be better but uncertainties in remotely sensed LAI can be huge. PLUMBER2 provides two independent LAI for this reason as at some sites LAI estimates are vastly different. I'm not sure providing an estimate from a single dataset is helpful or superior. If anything, it would have been valuable to include more LAI datasets to account for uncertainty and constrain these with site observations where available*

**Response 10:** Thank you for your comments regarding the uncertainties in remotely sensed LAI. We believe the value of using multiple LAI datasets to account for uncertainties in remotely sensed data. Here, we chose to utilize the reprocessed MODIS v6.1 LAI and use the site LAI to quantify its uncertainty. Please refer to response 2 for specific treatment. Please also see Comment 2 for more information.

***Comment 11 (L90):*** *Can the authors demonstrate that the data is smoother and more consistent? It is also not documented how this data was processed. Taking the raw values without additional quality control is rarely sufficient*

**Response 11:** Thank you for your suggestions. To provide a clearer introduction to the reprocessed MODIS LAI, we have added a summary of the processing methods along with the relevant citations. The

comparison before and after modification is as follows:

**Origin (L89):**

"And the reprocessed MODIS LAI is much smoother and more consistent with adjacent values, displaying better spatiotemporally continuous and consistency."

**Revised (L89):**

"The reprocessed MODIS LAI used the modified temporal spatial filter (mTSF) method for a simple data assimilation, then applied the post processing-TIMESAT (A software package to analyze time-series of satellite sensor data) Savitzky–Golay (SG) filter to get the final result. Site LAI validation shows that the reprocessed MODIS LAI is much smoother and more consistent with adjacent values than the original MODIS LAI, and closer to site observations (Lin et al., 2023; Yuan et al., 2011)."

*Comment 12 (L93): please check grammar*

**Response 12:** Thank you for your suggestions. We have revised it.

**Origin (L93):**

"LAI complements still use the reprocessed MODIS LAI. FVC complements use a global 300m PFT map"

**Revised (L93):**

"LAI filling still uses the reprocessed MODIS LAI, whereas the FVC filling employs a global 300 m PFT map"

*Comment 13 (L95): Were these PFT estimates cross-checked against site information e.g. from past papers? Global PFT datasets can be highly uncertain at flux towers even if provided at a high spatial resolution*

**Response 13:** We completely agree with your suggestion. Other reviewers have raised similar concerns, indicating the need for cross-checking of filled data to quantify their uncertainties.

Accordingly, we have provided the quantification of uncertainties for the filled data. The discrepancies between site data and filled data have been added to Sect. 3.2, illustrating the uncertainties of the filled data. The added part is as follows:

**Add (Sect. 3.2):**

[Figure]

**Figure 4.** Quantification of discrepancies between site data and filled data for (a) PCT_PFT, (b) maximum LAI, (c) canopy height, and (d) percentage of sand (at all sites for which both types of data are available). The 16 PFTs were divided into three main categories (bare soil, woody, and herbage) for separately quantification.

**Add (L299):**

Figure 4 quantifies the differences between site data and filled data at all sites for which both data sources are available, illustrating the inhomogeneities in the final dataset due to data filling. The differences in vegetation cover (including bare soil, woody, and herbaceous vegetation) generally fall within 20%, with a minority of sites exceeding 40%. The mean and median LAI differences are approximately 1 $m^2/m^2$. Canopy height deviations are primarily within 2 m, although a few sites exceed 4 m. Differences in sand content typically remain within 30%, with both mean and median differences below 15%. This quantification indicates that the filled data are generally reliable across most sites.

***Comment 14 (L99):*** *It is a little unclear how it is helpful providing soil type estimates using a dataset already applied in LSMs. As the authors state at the start, the use of such global datasets in flux site simulations risks discrepancies in model-obs comparisons*

**Response 14:** Thank you for your suggestions. We agree with you that global soil types do not offer obvious benefits in site-specific simulations. Therefore, the use of soil type estimates from these datasets is primarily to provide a reference. Additionally, following your recommendation, we have conducted cross-validation of these data (as detailed in Response 13), which enhances their reference value and reliability.

***Comment 15 (L104):*** *noting that elevation was provided in PLUMBER2*

**Response 15:** Thank you for your correction. Since the topography data was added later, we apologize for not having noticed this. We have removed the description of elevation.

**Origin (L106):**

"The topography attributes included elevation, slope, and aspect."

**Revised (L106):**

"The topography attributes included slope and aspect."

***Comment 16 (L109):*** *breakdown -> broken down*

**Response 16:** Thank you for your correction. We have revised it based on your suggestion.

**Origin (L109):**

"the FVC was further breakdown to different PFTs"

**Revised (L109):**

"the FVC was further broken down into different PFTs"

***Comment 17 (Figure 1):*** *"Data complement" is not entirely clear, do you mean supplementing site obs with global datasets?*

**Response 17:** Thank you for your question. We apologize for the ambiguity. "Data complement" indicates site measurements gap-filled with global datasets. we have changed the wording to express it more clearly.

**Origin (Figure 1):**

"Data complements"

**Revised (Figure 1):**

"Data filling"

***Comment 18 (L122):*** *Would be good to justify why (1) was done? On L133 you say non-consecutive years were kept to maximise utility of obs data, this somewhat contradicts that principle?*

**Response 18:** Thank you for your question. Observational data of only one year are generally unstable or unreliable. Consequently, we implemented step (1). Although subsequent screening resulted in some sites having only one year of data that meets the standards, this data is still considered stable. We have revised (1) to clarify this point.

**Origin (L122):**

"Sites with only one year of observations were excluded."

**Revised (L122):**

"Sites with only one year of observations were excluded for data stability and reliability."

***Comment 19 (L123):*** *As mentioned earlier, is this desirable, restricting how the data can be used for individual applications?*

**Response 19:** Thank you for your suggestions. We fully agree with your opinion. For individual research, this excluded data could also be very useful. Therefore, we have provided users with the opportunity to

access this data. Please refer to response 6 for specific treatment.

**Comment 20 (L126):** *ideally the VPD screening should be done in conjunction with temperature screening as both were used to convert VPD to specific humidity. A very good point that PLUMBER2 only used temperature but not VPD gapfilling information when screening specific humidity data*

**Response 20:** Thank you for your comment on the need to screen VPD. We are also grateful for your responsiveness to our questions in using PLUMBER2.

**Comment 21 (L127):** *These sites still provide non-corrected latent heat. It is not clear whether the EBF-corrected data is "better" (see https://egusphere.copernicus.org/preprints/2024/egusphere-2023-3084/)*

**Response 21:** Thank you for pointing out the availability of non-corrected latent heat. The non-closure of energy balance in eddy covariance (EC) flux tower observations has been a persistent issue and the subject of extensive discussion. Nevertheless, it is undeniable that the non-closure of the surface energy balance is one of the greatest challenges in quantifying the atmosphere-surface exchange of energy and water (Zhou et al., 2023).

The reasons for the energy imbalance are not attributable to a single factor. Different causes of imbalance may require different correction methods (Mauder et al., 2020). Overall, the Bowen ratio closure method demonstrates better performance compared to other closure methods (Zhou et al., 2023). Therefore, it can be said that applying the Bowen ratio for energy balance closure is generally advantageous in most cases.

We appreciate your thoughtful comments and hope these replies satisfy you.

**Comment 22 (L140):** *I don't quite follow this?*

**Response 22:** Thank you for your question. We appreciate the opportunity to clarify this point. In our study, we aimed to include as many site-observed data as possible. For sites where FVC data was missing, we used values close to the FVC as replacements, including the percentage of vegetation flux footprint contribution and dense forest canopy basal area. We treated them as FVC in this paper.

We have changed the wording to express it clearly. The comparison before and after modification is as follows:

**Origin (L140):**

"For sites lacking a direct FVC representation but providing information on the percentage of vegetation flux footprint contribution or dense forest canopy basal area"

**Revised (L140):**

"For sites lacking FVC data but providing the percentage of vegetation flux footprint contribution or dense forest canopy basal area"

***Comment 23 (L143):*** *please check grammar*

**Response 23:** Thank you for your correction. We have revised it.

**Origin (L143):**

"In the case of grassland and cropland sites, both surface cover landscapes are usually homogeneous cover and manual management."

**Revised (L143):**

"In the case of grassland and cropland sites, the vegetation cover type typically exhibits a high degree of homogeneity."

***Comment 24 (L148):*** *is this really true?*

**Response 24:** Thank you for your question. This approach does indeed involve some uncertainty. However, due to the lack of more detailed data, we adopted this simple assumption. A similar approach was used by Bonan et al. (2002), where they considered that bare ground might not be present even in semiarid regions with sparse, yet homogeneous land cover.

***Comment 25 (L150-156):*** *All of this needs further details, I don't follow how these steps were done*

**Response 25:** Thank you for your suggestions. We have added more detailed descriptions of these steps to ensure clear and comprehensible. We hope these revisions meet your satisfaction.

**Origin (L149):**

"For data completeness, we used the $PFT_{local}$ maps to complement the data for sites lacking site-observed vegetation cover proportion. After that, we further breakdown the FVC data in terms of different PFTs to align with the requirements of LSMs simulation using PFTs. First, trees and shrubs were classified as evergreen or deciduous, as well as coniferous or broadleaf types, based on the vegetation type expressed in the data sources. Next, Köppen-Geiger climate classification maps are employed to categorize the climate type of PFTs using the method proposed by Poulter et al. (2011). To better represent the C3 and C4 grasses, we prioritize segmentation based on the data source descriptions. If site description was not available, then segmentation was performed using the Still et al. (2003) method, which uses flux tower air temperature, precipitation, and reprocessed MODIS Version 6.1 LAI. "

**Revised (L149):**

"After that, trees and shrubs were classified as evergreen or deciduous, coniferous or broadleaf, based on the vegetation type. As an example, eucalyptus trees are classified as evergreen broadleaf trees. For data completeness, we used the $PFT_{local}$ maps to fill in data for sites lacking site-observed FVC.

We further break down the FVC into PFTs to meet the requirements of LSM simulations using PFTs. The breakdown method is as follows: First, the climate type of PFT was determined according to the Köppen climate classification (Poulter et al., 2011). Then, C3 and C4 grasses are partitioned using site descriptions. If site descriptions are unavailable, flux tower air temperature, precipitation, and the

reprocessed MODIS LAI are used to calculate LAI proportions under C3/C4 climatic conditions, to estimate the C3/C4 grass proportions (Still et al., 2003)."

**Comment 26 (L171):** *It would have been valuable to use these site-observed values to constrain remotely-sensed (MODIS) LAI, was this step done? It would provide a useful guide as to how reliable the MODIS estimates are*

**Response 26:** Thank you for your question. In Section 2.3, for the modeling assessment of attribute data, we scaled the MODIS LAI time series using site-observed maximum LAI value.

Based on your comments, we decided to add an explanation in Section 2.3 about using site data, detailing how these site attributes were applied in the simulations. The added information is as follows:

**Add (L225):**

"In the site data simulations, we scaled the default LAI time series using maximum LAI, corrected the default canopy height using site canopy height, and replaced the default topsoil texture (0-28.9 cm) with site soil texture. For sites with multiple PFTs, we calculated the LAI for each PFT using growing degree days and PCT_PFT (Lawrence and Chase, 2007). Canopy height was classified into three groups based on PFT (trees, shrubs, or grassland), with site data used to adjust the default values for the corresponding group, while the other two groups retained their default values."

**Comment 27 (L195):** *elevation was provided for each site in PLUMBER2 so how is this an advance?*

**Response 27:** Thank you for your question. The topographic data was initially not included in the attribute dataset but was added at a later stage. This resulted in our neglect of elevation in PLUMBER2. We apologize for this. The elevation data provided here is essentially consistent with that in PLUMBER2. We have revised the relevant descriptions accordingly.

**Origin (L194):**

"The topography data encompasses site elevation, slope and aspect. These data are gathered from site descriptions in literature, regional networks, FLUXNET and AmeriFlux BADM files. Specifically, we acquired elevation for 89 sites, slope for 57 sites, and aspect for 49 sites from these reference sources. In the AU-Lit site, where site elevation data was unavailable from the aforementioned references, we used the elevation given in Ukkola et al. (2022)."

**Revised (L194):**

"The topography data encompass slope and aspect measurements. These data are gathered from site descriptions in published literature, regional network, FLUXNET and AmeriFlux BADM files. Specifically, we acquired slope measurements for 57 sites, and aspect information for 49 sites from these reference sources."

***Comment 28 (L203):*** *This was also provided in PLUMBER2, would be interesting to know if the authors identified different heights to what was reported?*

**Response 28 (L203):** Thank you for your question. We have noted that PLUMBER2 provides reference measurement heights. Here, the attribute dataset provides the reference measurement height for wind speed. We observed that its values are generally consistent with the reference measurement heights in PLUMBER2 at most sites, with minimal differences. Additionally, following the suggestions of other reviewers, we have included the reference measurement heights for air temperature and humidity in the attribute dataset.

***Comment 29 (L225):*** *these are not really climate variables?*

**Response 29:** Thank you for your correction. these variables are indeed directly used when describing climatic conditions. We have revised it.

**Origin (L225):**

"The discrepancy of site data relative to default data is compared by an ensemble of climate-related variables, including…"

**Revised (L225):**

"The discrepancy of site data relative to default data is compared by variables related to land surface energy, water, and photosynthesis processes, including…"

***Comment 30 (L227):*** *Runoff is not available at flux sites so I don't follow how it was used?*

**Response 30:** Thank you for your question. Yes, there is a lack of observations of TR and a limitation in the observational depth of SWC.As mentioned in Eq (1), for SWC and TR, we did not use observational data but relied on simulation results to show their differences.

***Comment 31 (L229):*** *grammar*

**Response 31:** Thank you for your correction.

**Origin (L228):**

"To quantify differences between output from the site and default data, and considering the seasonal fluctuations in the impacts of soil and vegetation on climate-related variables (Dirmeyer, 2011; Forzieri et al., 2020). We designed a statistical indicator called the percentage of mean difference (MD %) (Eq. 1), which is calculated as"

**Revised (L228):**

"To quantify the differences between the output from the site and default data, while also accounting for seasonal fluctuations in the impacts of soil and vegetation on simulated variables (Dirmeyer, 2011; Forzieri et al., 2020), we designed a statistical indicator called the percentage of mean

difference (MD %) (Eq. 1). The indicator is calculated as"

**Comment 32 (L257):** *I don't see superscript "e" in the table?*

**Response 32:** Thank you for your question. The "e" in the "LAI_default" column for the "US-KS2" row specifies the particular year of the maximum LAI.

**Comment 33 (Figure2):** *The IGBP type was provided in PLUMBER2, would be interesting to know how different the PFTs provided here are? Much of the PF T information in PLUMBER2 came from site-specific data provided on Fluxnet and regional network websites*

**Response 33:** Thank you for your question. In this study, the PFT data is sourced from site-related literature, regional networks, and FLUXNET BADM files. We applied certain approximations and processing to the original vegetation cover data, categorizing it according to the 16 PFTs classification to facilitate its use in land surface models. Additionally, these data provide reference sources to allow users to access the original data.

**Comment 34 (L285):** *This is why it would have been useful to include alternative LAI products and select the most suitable one at each site (PLUMBER2 attempted this but this could no doubt be improved). Only relying on one dataset is arguably not an improvement given the discrepancies*

**Response 34:** Thank you for your comment. Undoubtedly, providing more LAI products would help readers understand its uncertainties. In our study, we used site data to quantify the uncertainty of remotely sensed LAI (Response 13). Please refer to Response 2 for specific treatment.

**Comment 35 (L292):** *"provides" might be better*

**Response 35:** Thank you for your correction and suggestion. We have rephrased this sentence to make it clearer. The comparison before and after modification is as follows:

**Origin (L292):**

"For the maximum LAI, the file furnishes the range of years for maximum LAI, and the maximum for a specific year."

**Revised (L292):**

"For the maximum LAI, the file provides both the year range over which the maximum LAI was observed and the specific maximum value for a given year"

**Comment 36 (L355):** *Very much agree with this statement. Would be great for this paper to call for the provision of these data in flux data releases*

**Response 36:** We fully agree with you. We have reformulated this sentence. The comparison before and after modification is as follows:

**Origin (L355):**

"In land surface community, flux tower attribute data is currently not given enough attention."

**Revised (L355):**

"In land surface research community, flux tower attribute data is currently not given enough attention. However, the site attribute data is almost as important as the flux tower observations themselves. We recommend that future flux tower datasets, such as the successors of FLUXNET2015, provide standardized site attributes."

**Comment 37 (L356):** *This is with the caveat that not all sites had site-observed values for the attributes provided?*

**Response 37:** Thank you for your suggestions. We acknowledge this limitation and have clarified it in the manuscript. The comparison before and after modification is as follows:

**Origin (L355):**

"Here, we have acquired 90 sites with dependable quality by comprehensive selection, and provided data on vegetation, soil, and topography attributes observed at the site."

**Revised (L355):**

"In this study, we have acquired 90 sites with high quality by a comprehensive selection process, and provided as many site-observed data as possible on vegetation, soil, and topography attributes."

**Comment 38 (L362):** *grammar*

**Response 38:** Thank you for your correction. We have revised it. The comparison before and after modification is as follows:

**Origin (L362):**

"These updates will therefore help the model's evolution. To collect more site-observed attribute data, while taking into account the diversity described within the same attribute data, particularly the percentage of vegetation cover. We made a few approximations and assumptions in the data collection procedure."

**Revised (L362):**

"Therefore, these updates will help the model's evolution. To collect more site-observed attribute data, while considering the diversity described within the same attribute data, particularly the percentage of vegetation cover, we made a few approximations and assumptions in the data collection procedure."

***Comment 39 (L364):*** *please provide examples here of what you mean*

**Response 39:** Thank you for your suggestions. We have revised it.

**Origin (L364):**

"we made a few approximations and assumptions in the data collection procedure."

**Revised (L364):**

"we made a few approximations and assumptions in the data collection procedure, such as using approximation substitution and site pictures to assist in judgment."

***Comment 40 (L372-379):*** *this section could be clearer*

**Response 40:** Thank you for your comment. We appreciate your suggestion to clarify this section. We have revised the text to improve clarity.

Considering your and other reviewers' comments on the discussion of LAI variations, we believe the modeling assessment of attribute data has primarily focused on the magnitude of the impact of attribute data, rather than on sensitivity analyses. We recognize that this passage may cause some misunderstanding. Therefore, after careful consideration, we removed the argument from the manuscript.

The comparison before and after modification is as follows:

**Delete (L376):** "Notably, unit LAI variations elicit more substantial fluctuations in fluxes at lower LAI values (usually less than 2 $m^2/m^2$), according to Launiainen et al. (2016). In light of that, all of the sites we chose have LAI values greater than 2 $m^2/m^2$, except US-GLE, the impact of LAI obtained here are relatively minor."

**Origin (L371):**

"According to the results, which are in line with earlier research (Dai et al., 2019a), vegetation cover appreciably affects each of the eight variables examined. And among the four attributes, net radiation was the most affected by vegetation cover (Fig. 5). This is due to the cover of plants being the most noticeable surface feature, directly changing surface energy absorption. The net radiation simulation was enhanced using the site PCT_PFT, but the latent and sensible heat did not perform as well. This may be related to the model's previous development and evaluation, which was mostly centered on the IGBP classifications (Dai et al., 2019b; Zhang et al., 2017; Zhu et al., 2017)."

**Revised (L371):**

"The results are in line with previous research (Dai et al., 2019a), showing that vegetation cover appreciably affects each of the eight variables examined, often being the dominant attribute (Figure 5). This is due to the cover of plants being the most noticeable surface feature, directly altering surface energy absorption. The net radiation simulation was improved using the site PCT_PFT, but the performance of latent and sensible heat simulations was suboptimal. This may be related to uncertainties in the model itself as well as other input data. Such as the vegetation biophysical parameters, soil thermal and hydraulic conductivities, etc. (Dai et al., 2019b; Zhang et al., 2017; Zhu et al., 2017)."

***Comment 41 (392):*** *what do you mean by "the full realization of differences in soil infiltration capacity "?*

**Response 41:** Thank you for your question. We appreciate the opportunity to clarify this point. By "the full realization of differences in soil infiltration capacity," we mean that during periods of high precipitation intensity, the distinct infiltration capacities of different soil textures become more evident and impactful. In other words, soils with different textures exhibit varying abilities to absorb and transmit water, which becomes particularly pronounced under heavy rainfall conditions. We have revised the manuscript to make this clearer. The comparison before and after modification is as follows:

**Origin (L391):**

"This is partly attributed to increased water availability and largely to the full realization of differences in soil infiltration capacity under high-intensity precipitation."

**Revised (L391):**

"This is partly attributed to increased water availability and largely to the pronounced differences in soil infiltration capacity under high-intensity precipitation cases."

***Comment 42 (L399):*** *I don't follow this? "Nevertheless, the data sources were published works, leading to deficiencies for certain sites "*

**Response 42:** Thank you for your question. What we intended to convey is that while we combined multiple sources to collect as much site-observed attribute data as possible, our reliance on published works meant that some sites had incomplete data. We have revised the manuscript to make this clearer.

**Origin (L399):**

"Nevertheless, the data sources were published works, leading to deficiencies for certain sites. And the attribute data we collected focused on fundamental soil and vegetation information."

**Revised (L399):**

"Nevertheless, the data sources we collected were primarily from published works, which led to some missing data for certain sites. And the attribute data focused only on soil and vegetation information."

***Comment 43 (L407):*** *"facilitating perception of the authentic feedback with diverse schemes and processes." What does this mean?*

**Response 43:** Thank you for your comment. By "facilitating perception of the authentic feedback with diverse schemes and processes," we mean that using site-observed attribute data allowed us to better understand the true effects of different schemes and processes. We have revised the manuscript to make this clearer.

**Origin (L406):**

"Working with site-observed attribute data enabled us to narrow down factors contributing to model uncertainties, facilitating perception of the authentic feedback with diverse schemes and processes."

**Revised (L406):**

"Working with site-observed attribute data enabled us to narrow down factors contributing to model uncertainties, thereby enhancing our understanding of the true effects of diverse schemes and processes."

***Comment 44 (L441):*** *This would be a great place to call for attribute data to be routinely released as part of flux tower data collections so ancillary data could be accessed more easily and routinely*

**Response 44:** Thank you for your valuable suggestion. We totally agree with you. We have revised the manuscript to include this important point.

**Added (L442):**

"We strongly advocate for the routine release of attribute data as part of flux tower data. Making such ancillary data more easily and routinely accessible would greatly increase the value and usability of the data."

We would like to thank you for your professional review work, constructive comments, and valuable suggestions on our manuscript. We hope the correction made will meet with approval. These comments and suggestions have significantly improved the quality of our manuscript. Once again, thank you very much for the comments and suggestions.

**References:**

Bonan, G. B., Levis, S., Kergoat, L., and Oleson, K. W.: Landscapes as patches of plant functional types: An integrating concept for climate and ecosystem models: PLANT FUNCTIONAL TYPES AND CLIMATE MODELS, Global Biogeochem. Cycles, 16, 5-1-5–23, https://doi.org/10.1029/2000GB001360, 2002.

Dai, Y., Yuan, H., Xin, Q., Wang, D., Shangguan, W., Zhang, S., Liu, S., and Wei, N.: Different representations of canopy structure—A large source of uncertainty in global land surface modeling, Agricultural and Forest Meteorology, 269–270, 119–135, https://doi.org/10.1016/j.agrformet.2019.02.006, 2019a.

Dai, Y., Wei, N., Yuan, H., Zhang, S., Shangguan, W., Liu, S., Lu, X., and Xin, Y.: Evaluation of Soil Thermal Conductivity Schemes for Use in Land Surface Modeling, J Adv Model Earth Syst, 11, 3454–3473, https://doi.org/10.1029/2019MS001723, 2019b.

Launiainen, S., Katul, G. G., Kolari, P., Lindroth, A., Lohila, A., Aurela, M., Varlagin, A., Grelle, A., and Vesala, T.: Do the energy fluxes and surface conductance of boreal coniferous forests in Europe

scale with leaf area?, Glob Change Biol, 22, 4096–4113, https://doi.org/10.1111/gcb.13497, 2016.

Lawrence, P. J. and Chase, T. N.: Representing a new MODIS consistent land surface in the Community Land Model (CLM 3.0), J. Geophys. Res., 112, G01023, https://doi.org/10.1029/2006JG000168, 2007.

Lin, W., Yuan, H., Dong, W., Zhang, S., Liu, S., Wei, N., Lu, X., Wei, Z., Hu, Y., and Dai, Y.: Reprocessed MODIS Version 6.1 Leaf Area Index Dataset and Its Evaluation for Land Surface and Climate Modeling, Remote Sensing, 15, 1780, https://doi.org/10.3390/rs15071780, 2023.

Mauder, M., Foken, T., and Cuxart, J.: Surface-Energy-Balance Closure over Land: A Review, Boundary-Layer Meteorol, 177, 395–426, https://doi.org/10.1007/s10546-020-00529-6, 2020.

Poulter, B., Ciais, P., Hodson, E., Lischke, H., Maignan, F., Plummer, S., and Zimmermann, N. E.: Plant functional type mapping for earth system models, Geosci. Model Dev., 4, 993–1010, https://doi.org/10.5194/gmd-4-993-2011, 2011.

Still, C. J., Berry, J. A., Collatz, G. J., and DeFries, R. S.: Global distribution of $C_3$ and $C_4$ vegetation: Carbon cycle implications: $C_4$ PLANTS AND CARBON CYCLE, Global Biogeochem. Cycles, 17, 6-1-6–14, https://doi.org/10.1029/2001GB001807, 2003.

Ukkola, A. M., Abramowitz, G., and De Kauwe, M. G.: A flux tower dataset tailored for land model evaluation, Earth Syst. Sci. Data, 14, 449–461, https://doi.org/10.5194/essd-14-449-2022, 2022.

Yuan, H., Dai, Y., Xiao, Z., Ji, D., and Shangguan, W.: Reprocessing the MODIS Leaf Area Index products for land surface and climate modelling, Remote Sensing of Environment, 115, 1171–1187, https://doi.org/10.1016/j.rse.2011.01.001, 2011.

Zhang, X., Dai, Y., Cui, H., Dickinson, R. E., Zhu, S., Wei, N., Yan, B., Yuan, H., Shangguan, W., Wang, L., and Fu, W.: Evaluating common land model energy fluxes using FLUXNET data, Adv. Atmos. Sci., 34, 1035–1046, https://doi.org/10.1007/s00376-017-6251-y, 2017.

Zhou, Y., Sühring, M., and Li, X.: Evaluation of energy balance closure adjustment and imbalance prediction methods in the convective boundary layer – A large eddy simulation study, Agricultural and Forest Meteorology, 333, 109382, https://doi.org/10.1016/j.agrformet.2023.109382, 2023.

Zhu, S., Chen, H., Zhang, X., Wei, N., Shangguan, W., Yuan, H., Zhang, S., Wang, L., Zhou, L., and Dai, Y.: Incorporating root hydraulic redistribution and compensatory water uptake in the Common Land Model: Effects on site level and global land modeling, J. Geophys. Res. Atmos., 122, 7308–7322, https://doi.org/10.1002/2016JD025744, 2017.

---

## Author Comment (AC4)

**Reply to Referee #4's comments**

**Title: A flux tower site attribute dataset intended for land surface modeling**
**No.: essd-2024-77**

*Shi et al. present a study on a flux tower site attribute dataset intended for land surface modeling. This dataset is very valuable for land surface modeling and beyond. Overall, the manuscript is well-written, I have some additional comments for consideration.*

Thank you for your careful evaluation of this manuscript. We greatly appreciate your positive and constructive comments on our manuscript, which have significantly improved the quality of our manuscript.

All comments are addressed on a point-by-point basis below. The comments are laid out below in italicized font and specific concerns are numbered. Our response is given in normal font. The list of all related changes is given in blue text.

**Comment 1 (L102):** *"Picking years with a low gap-filled percentage for fluxes (latent and sensible heat) and vapor pressure deficit (VPD)." Why were these variables chosen for picking? Why not include precipitation, temperature, etc.? Please provide an explanation.*

**Response1:** Thank you for your questions. The quality screening in this study is based on the PLUMBER2 dataset. PLUMBER2 has already screened meteorological data, including the five key variables that have the largest influence on LSM simulations: incoming shortwave radiation, precipitation, air temperature, air humidity, and wind speed. Therefore, we did not conduct additional screening for variables such as precipitation and air temperature. We have added this information to the manuscript.

The reason for screening for VPD is that air humidity is calculated using VPD. However, the screening process of PLUMBER2 did not consider the gap-filled situation of VPD.

We selected latent heat and sensible heat for two main reasons: (1) latent and sensible heat are two of the most critical variables that need to be assessed in land-atmosphere exchange. Consequently, when the quality of latent and sensible heat is poor, we exclude all fluxes. (2) Lower quality observations for latent and sensible heat usually indicate reduced quality of other flux exchanges (e.g., carbon exchanges including GPP and respiration; friction velocities).

**Origin (L177):**

"The PLUMBER2 dataset got 170 sites by screening meteorological data."

**Revised (L117):**

"The PLUMBER2 dataset got 170 sites by screening meteorological data (including five key variables that have the largest influence on LSM simulations: incoming shortwave radiation, precipitation, air temperature, air humidity, and wind speed.)."

**Comment 2:** *One of our studies (https://papers.ssrn.com/sol3/papers.cfm?abstract_id=4732309) conducted land surface model evaluations at the site level using the PLUMBER2 datasets, which are very valuable. We took time to find seek additional model inputs, such as soil texture information, from the literature. This study is convenient for land surface modelers. Therefore, it would be even more valuable if it included more sites and longer period of data beyond the PLUMBER2 sites.*

**Response 2:** We sincerely appreciate your valuable suggestions and recognition. PLUMBER2 includes the major datasets available since the initial release of flux tower data. However, since flux tower datasets like FLUXNET2015 have not been released new versions, PLUMBER2 does not contain observational data updated in recent years. We acknowledge that these data are far from sufficient, but we believe this is a good starting point.

Based on your suggestions and those of other reviewers, we have included a call for the release of flux tower attribute data in the manuscript. The revisions are as follows:

**Origin (L355):**

"In land surface community, flux tower attribute data is currently not given enough attention."

**Revised (L355):**

"In land surface community, flux tower attribute data is currently not given enough attention. However, the site attribute data is almost as important as the flux tower observations themselves. We hope that future flux tower datasets will provide standardized site attributes."

**Added (L442):**

"We strongly advocate for the routine release of attribute data as part of flux tower data. Making such ancillary data more easily and routinely accessible would greatly increase the value and usability of the data."

**Comment 3:** *Another study of ours developed global 1km land surface parameters for earth system modeling (https://essd.copernicus.org/articles/16/2007/2024/essd-16-2007-2024.html), sharing some data sources with this study, such as PFTs classification. Combining this 1km data with the site-level study could enhance land surface modeling. For example, global 1km data could provide topography attributes for sites lacking this information. This could be discussed in the manuscript.*

**Response 3:** Thank you very much for your suggestions. We have considered using global topography data to supplement site-observed slope and aspect. However, given that most sites are located on flat terrain, as indicated by both site data and global data, and considering the potential for scale mismatch,

we believe that adding this data may offer limited benefit without a detailed assessment. Moreover, we observed certain discrepancies between global slope and site slope at some sites, as shown in the table below. Therefore, Therefore, we have not added global terrain data currently.

We appreciate your thoughtful comments and hope these replies satisfy you.

| site | slope_global | slope_site |
|------|-------------|-----------|
| AT-Neu | 15.7% | Flat |
| CH-Fru | 5.2% | < 5% |
| CN-HaM | 4.5% | Flat |
| DE-Bay | 3.1% | 10-15% |
| DE-Obe | 3.3% | 5-10% |
| FR-Pue | 3.0% | Flat |
| IT-MBo | 7.2% | Flat |
| US-Me4 | 9.4% | 5-10% |
| US-NR1 | 4.9% | 5-10% |
| US-SRG | 3.6% | < 2% |

**Comment 4 (L92):** *"Three global datasets were used to complement attribute data of sites lacking site-observed FVC, LAI, and soil texture." It's great to analyze the uncertainty by using global datasets to fill site data. How accurate are these global datasets? The authors could analyze the consistency between sites with complete attributes and the corresponding global datasets. This information would be valuable for readers to understand the uncertainties introduced by using global datasets.*

**Response 4:** We fully agree with your suggestion. Other reviewers have raised similar concerns requiring cross-checking of filled data to quantify their uncertainties.

We provide the quantification of uncertainties in the final dataset resulting from data filling. After careful consideration, the quantification of discrepancies between site data and filled data has been added to Sect. 3.2, illustrating the uncertainties of the filled data. The added information is as follows.

**Add (Sect. 3.2):**

[Figure]

**Figure 4.** Quantification of discrepancies between site data and filled data for (a) PCT_PFT, (b) maximum LAI, (c) canopy height, and (d) the percentage of sand (at all sites for which both types of data sources are available). The 16 PFTs were divided into three main categories (bare soil, woody, and herbage) to be quantified separately quantification.

**Add (L299):**

Figure 4 quantifies the differences between site data and filled data at all sites for which both data sources are available, illustrating the inhomogeneities in the final dataset due to data filling. The differences in vegetation cover (including bare soil, woody, and herbaceous vegetation) generally fall within 20%, with a minority of sites exceeding 40%. The mean and median LAI differences are approximately 1 $m^2/m^2$. Canopy height deviations are primarily within 2 m, although a few sites exceed 4 m. Differences in sand content typically remain within 30%, with both mean and median differences below 15%. The quantification indicates that the filled data are generally reliable across most sites.

*Comment 5 (L195): It seems not all sites have elevation, slope, and aspect information from literature. How were these attributes obtained for sites lacking them? Seems not mentioned in the manuscript.*

**Response 5:** Thank you for your question. Yes, we did not find suitable global topographic data and only provided site-observed slope and aspect information in the attribute dataset.

*Comment 6 (Table 2): Should "LAI_default" be "LAI_max_default"? Similarly, for "LAI_site."*

**Response 6:** Thank you for your correction. This should be "LAI_max_default". We have revised the relevant description.

*Comment 7 (L220): "Resulting in six available sites. These sites were simulated to show the respective impact of different attributes in model results." This sentence is unclear. How many experiments were run for each site? This section could be written more clearly to make Section 3.3 easier to understand.*

**Response 7:** Thank you for your suggestion. We fully agree with you. We have provided a description of the experiments we conducted. The comparison before and after modification is as follows:

**Origin (L219):**

"These sites were simulated to show the respective impact of different attributes in model results. Table 2 provides an overview of the chosen sites along with their corresponding attribute information."

**Revised (L219):**

"Table 2 provides an overview of the selected sites along with their corresponding attribute information. Each site was simulated three times: 1) using site data for each attribute at each site, 2) using default data for each attribute at each site, and 3) using default data for the corresponding attribute at sites selected for each attribute separately, while maintaining site data for the remaining attributes. The comparison between simulations (1) and (3) aims to demonstrate the individual impact of each attribute, while the comparison between simulations (1) and (2) shows the combined impact of all four attributes."

We would like to thank you for your professional review work, constructive comments, and valuable

suggestions on our manuscript. We hope the correction made will meet with approval. These comments and suggestions have significantly improved the quality of our manuscript. Once again, thank you very much for the comments and suggestions.

---

## Author Response (AR2)

**Author's response**

**Title: A flux tower site attribute dataset intended for land surface modeling**
**No.: essd-2024-77**

We sincerely thank the editor and all reviewers for their efforts. It is a great pleasure to have our work recognized. We sincerely thank the topic editor for pointing out our shortcomings. We have made changes in response to these comments.

The comments are presented in the italicized font and specific concerns are numbered. Our response is given in normal font. The list of all related changes is provided in blue text.

**Comment 1:** *Figure 4: Please explain the symbols in the boxplot in the caption.*

**Response 1:** Thank you for your careful examination. We have revised it.

**Add (Figure 4 caption):**

"Boxes (25th and 75th percentiles) and whiskers (5th and 95th percentiles), with median (blue line) and mean (blue triangle). Hollow circles denote outliers defined as values greater than 1.5 times the interquartile range from the nearest 25th or 75th percentile."

**Comment 2:** *Figures 5 and 7: Please avoid the use of green-red pair.*

**Response 2:** Thank you for your correction. We have used a new color scheme that is colorblind friendly.

**Origin (Figure 5):**

[Figure]

**Revised ((Figure 5):**

[Figure]

**Origin (Figure 7):**

[Figure]

**Revised (Figure 7):**

[Figure]

These updates encompass all the revisions we have made. We are sincerely grateful for the time and effort that the topic editor dedicated to providing feedback on our manuscript.

We appreciate the Editors᾽ warm work earnestly, and hope the correction made will meet with approval.